

# A novel method for calculating ambient aerosol
# liquid water contents based on measurements of a
# humidified nephelometer system
**Ye Kuang[1,2], ChunSheng Zhao[1], Gang Zhao[1], JiangChuan Tao[1,2], Nan Ma[2], YuXuan Bian[3]**
[1]{Department of Atmospheric and Oceanic Sciences, School of Physics, Peking University,
Beijing, China}
[2]{Institute for Environmental and Climate Research, Jinan University, Guangzhou 511443,
China}
[3]{State Key Laboratory of Severe Weather, Chinese Academy of Meteorological Sciences}
Correspondence to: C. S. Zhao (zcs@pku.edu.cn)
**Abstract**
Water condensed on ambient aerosol particles plays significant roles in atmospheric
environment, atmospheric chemistry and climate. So far, no instruments are available for real-time
monitoring of ambient aerosol liquid water contents (ALWC). In this paper, a novel method is
proposed to calculate ambient ALWC based on measurements of a three-wavelength humidified



nephelometer system. A humidified nephelometer system measures aerosol light scattering
coefficients and backscattering coefficients at three wavelengths under dry and different relative
humidity (RH) conditions, and therefore provides measurements of light scattering enhancement
factor $f(\text{RH})$. The proposed method of calculating ALWC includes two steps. The first step is
estimating total volume concentration of ambient aerosol particles in dry state ($V_a(\text{dry})$) with a
machine learning method based on measurements of the "dry" nephelometer. The estimated
$V_a(\text{dry})$ agrees well with the measured $V_a(\text{dry})$. The second step is estimating the volume growth
factor Vg(RH) of ambient aerosol particles due to water uptake using $f(\text{RH})$ and Ångström
exponent. The ALWC is calculated from the estimated $V_a(\text{dry})$ and Vg(RH). Uncertainty analysis
of the estimated $V_a(\text{dry})$ and Vg(RH) is conducted. This research have bridged the gap between
$f(\text{RH})$ and Vg(RH). The advantage of this new method is that the ambient ALWC can be obtained
using only measurements from a three-wavelength humidified nephelometer system. This method
will facilitate the real-time monitoring of the ambient ALWC and help for studying roles of aerosol
liquid water in atmospheric chemistry, secondary aerosol formation and climate change.
**1. Introduction**

Atmospheric aerosol particles play significant roles in atmospheric environment, climate,

human health and the hydrological cycle, and have received much attention in recent decades.
One of the most important constituents of ambient atmospheric aerosol is liquid water. The
content of condensed water on ambient aerosol particles depends mostly on both aerosol
hygroscopicity and ambient relative humidity (RH). Results of previous studies demonstrate that
liquid water contributes greatly to the total mass of ambient aerosol particles when the ambient


RH is higher than 60% (Bian et al., 2014). And aerosol liquid water has large impacts on aerosol
optical properties and aerosol radiative effects (Tao et al., 2014;Kuang et al., 2016). Condensed
liquid water on aerosol particles can also serve as a site for multiphase reactions which perturb
local chemistry and also further influence the aging processes of aerosol particles (Martin, 2000).
Recent studies have shown that aerosol liquid water serves as a reactor which help for efficiently
transforming sulfur dioxide to sulfate during haze events and plays crucial roles in worsening
atmospheric environment on the North China Plain (NCP) (Wang et al., 2016;Cheng et al.,
2016). Hence, the real-time monitoring of ambient aerosol liquid water content (ALWC) is of
crucial importance to gain more insights into the roles of aerosol liquid water in atmospheric
chemistry, aerosol aging processes and aerosol optical properties.
Few techniques are available for measuring ALWC. The humidified tandem differential
mobility analyser systems (HTDMAs) are useful tools and widely used to measure hygroscopic
growth factors of ambient aerosol particles (Rader and McMurry, 1986;Wu et al., 2016;Meier et
al., 2009). Hygroscopicity parameters retrieved from measurements of HTDMAs can be used to
calculate volumes of liquid water. Nevertheless, HTDMAs can not be used to measure the total
aerosol water volume because they are not capable of measuring the hygroscopic properties of
the entire aerosol size distribution. With size distributions of aerosol particles in their ambient
state and dry state, the aerosol water volume can de estimated. Engelhart et al. (2011) deployed a
Dry-Ambient Aerosol Size Spectrometer to measure the aerosol liquid water content and volume
growth factor of fine particulate matter. This system provides only aerosol water content of
aerosol particles within certain size range ( particle diameter less than 500 nm for setup of
Engelhart et al. (2011)). In addition, in conjunction with aerosol thermodynamic equilibrium
models, ALWC can also be estimated with detailed aerosol chemical information. However,





simulations of aerosol hygroscopicity and phase state by using thermodynamic equilibrium
models are still very complicated even under the thermodynamic equilibrium hypothesis and
these models may cause large bias when used for estimating ALWC (Bian et al., 2014).

A humidified nephelometer system which measures aerosol light scattering coefficient ($\sigma_{sp}$)

under dry and different RH conditions is a relatively early method proposed for studying aerosol
hygroscopicity (Covert et al., 1972). It provides information about aerosol light scattering
enhancement factor $f(\mathrm{RH})$ . One advantage of this method is that it has a fast response time and
thus measurements can be made continuously which facilitates the monitoring of changing
ambient conditions. Another advantage of this method is that it provides information about the
overall aerosol hygroscopicity of the entire aerosol size distribution. Both measured $\sigma_{sp}$ of
aerosol particles in dry state and $f(\mathrm{RH})$ vary strongly with parameters of particle number size
distribution (PNSD), it is difficult to directly link them with volume of aerosol particles in dry
state ($V_a(\mathrm{dry})$) and the volume growth factor Vg(RH) of the entire aerosol population. So far, the
ALWC can not be directly estimated with measurements from only a humidified nephelometer
system. Several studies have shown that if PNSDs in dry state are measured, then an iterative
algorithm with Mie theory can be used to calculate an overall aerosol hygroscopic growth factor
g(RH)  based on measurements of $f(\mathrm{RH})$ (Zieger et al., 2010;Fierz-Schmidhauser et al., 2010).
In this iterative algorithm, the g(RH) is assumed to be independent of the aerosol diameter. Then
ALWC at different RH points can be calculated based on derived g(RH) and the measured
PNSD. This method not only requires additional measurements about PNSD, but also may result
in significant deviations of the estimated ALWC because that  g(RH) should be a function of
aerosol diameter rather than a constant value. In this paper, we proposed a novel method to
calculate the ALWC based only on measurements of a humidified nephelometer system.



## 2. Materials and methods

### 2.1 Datasets

Datasets from five field campaigns are used in this paper. Details about these field

campaigns (campaigns F1 to F5, see Table S1) are introduced in the supplemental material.

During these field campaigns, sampled aerosol particles have aerodynamic diameters less than

10 µm (by passing through an impactor). The PNSDs in dry state which range from 3nm to

10µm were jointly measured by a Twin Differential Mobility Particle Sizer (TDMPS, Leibniz-

Institute for Tropospheric Research , Germany; Birmili et al. (1999)) or a scanning mobility

particle size spectrometer (SMPS) and an Aerodynamic Particle Sizer (APS, TSI Inc., Model

3321) with a temporal resolution of 10 minutes.. The mass concentrations of black carbon (BC)

were measured using a Multi-Angle Absorption Photometer (MAAP Model 5012, Thermo, Inc.,

Waltham, MA USA) with a temporal resolution of 1 minute. The aerosol light scattering

coefficients ($\sigma_{sp}$) at three wavelengths (450 nm, 550 nm, and 700 nm) were measured using a

TSI 3563 nephelometer (Anderson and Ogren, 1998).

Datasets about PNSD, BC and $\sigma_{sp}$ from campaigns F1 to F4 are referred to as D1. Datasets

about PNSD, BC and $\sigma_{sp}$ from campaigns F2, F4 and F5 are referred to as D2. Measurements of

PNSD and measurements from the humidified nephelometer system during campaign F5

(Wangdu campaign) are used to verify the proposed method of estimating $V_a(\text{dry})$ and calculate

the ambient ALWC. Details about the humidified nephelometer system during Wangdu

campaign are introduced in the supplemental material.

### 2.2 Mie theory



The first goal of this research is estimating $V_a(\text{dry})$ from $\sigma_{sp}$ measurements. The $V_a(\text{dry})$
can be integrated from measured PNSD. Thus, datasets of $\sigma_{sp}$ and PNSD are needed to
investigate relationships between $\sigma_{sp}$ and $V_a(\text{dry})$. To make sure the data quality of used datasets
of $\sigma_{sp}$ and PNSD, the closure between measured $\sigma_{sp}$ and $\sigma_{sp}$ calculated based on measured
PNSD and BC with Mie theory (Bohren and Huffman, 2008) is first investigated. Measured $\sigma_{sp}$
has problems regarding angular truncation errors and nonideality of light source. In order to
make sure the consistency between measured and modelled $\sigma_{sp}$, modelled $\sigma_{sp}$ are calculated
according to practical angular situations of the nephelometer (Anderson et al., 1996). Moreover,
during processes of modelling $\sigma_{sp}$, BC is considered to be half externally and half coreshell
mixed with other aerosol components, and the mass size distribution of BC used in Ma et al.
(2012) which was observed on the NCP is used in this research to account for the mass
distributions of BC at different particle sizes. The used refractive index and density of BC are
$1.80 - 0.54i$ and $1.5\text{g } cm^{-3}$ (Kuang et al., 2015). Used refractive index of non light-absorbing
aerosol components (other than BC) is$1.53 - 10^{-7}i$ (Wex et al., 2002). Calculation details based
on  the Mie theory please refer to Kuang et al. (2015). Datasets about PNSD and $\sigma_{sp}$ from D1 are
used to perform the closure investigation. Finally, during processes of investigating relationships
between $\sigma_{sp}$ and $V_a(\text{dry})$, data points in D1 with relative differences between measured $\sigma_{sp}$ at
550 nm and modelled $\sigma_{sp}$ at 550 nm greater than 10% are excluded. 10% is chosen because of
that measured PNSD has uncertainty of larger than 10% (Wiedensohler et al., 2012), and
measured $\sigma_{sp}$ has uncertainty of about 9%, this threshold can make sure that most used data
points are measured when instruments operated well. .
**2.3 κ-Köhler theory(Wiedensohler et al., 2012)**





To simulate the relationships between $f(\text{RH})$ and Vg(RH), κ-Köhler theory is used to
describe the hygroscopic growth of aerosol particles with different sizes, and the formula
expression of κ-Köhler theory can be written as follows (Petters and Kreidenweis, 2007):
$$\text{RH} = \frac{D^3 - D_d{}^3}{D^3 - D_d{}^3(1-\kappa)} \cdot \exp\left(\frac{4\sigma_{s/a} \cdot M_{water}}{R \cdot T \cdot D_p \cdot g \cdot \rho_w}\right) \qquad (1)$$
where D is the diameter of the droplet, $D_d$ is the dry diameter, $\sigma_{s/a}$ is the surface tension of
solution/air interface, T is the temperature, $M_{water}$ is the molecular weight of water, R is the
universal gas constant, $\rho_w$ is the density of water, and $\kappa$ is the hygroscopicity parameter. By
combining the Mie theory and the κ-Köhler theory, both $f(\text{RH})$ and Vg(RH) can be simulated.
In the processes of calculations for modelling $f(\text{RH})$ and Vg(RH), the treatment of BC is same
with those introduced in Sect.2.2. As aerosol particle grow due to aerosol water uptake, the
refractive index will change. In the Mie calculation, impacts of aerosol liquid water on the
refractive index are considered on the basis of volume mixing rule. The used refractive index of
liquid water is $1.33 - 10^{-7}i$ (Seinfeld and Pandis, 2006).
**2.4 Parameterization scheme for $f(\text{RH})$**
The $f(\text{RH})$ is defined as $f(\text{RH}) = \sigma_{sp}(RH, 550\,nm)/\sigma_{sp}(dry, 550\,nm)$ where
$\sigma_{sp}(RH, 550\,nm)$ and $\sigma_{sp}(dry, 550\,nm)$ represents $\sigma_{sp}$ at wavelength 550 nm under certain
RH and dry conditions. Additionaly, Vg(RH) is defined as $Vg(RH) = V_a(RH)/V_a(dry)$, where
$V_a(RH)$ represents total volume of aerosol particles under certain RH conditions.
A physically based single-parameter representation is proposed by Brock et al. (2016) to
describe $f(\text{RH})$. The parameterization scheme is written as:




$$f(\text{RH}) = 1 + \kappa_{sca}\frac{RH}{100-RH} \quad (2)$$
where $\kappa_{sca}$ is the parameter which fits $f(\text{RH})$ best. Here, a brief introduction is given about the
physical understanding of this parameterization scheme. For aerosol particles whose diameters
larger than 100 nm, regardless of the kelvin effect, the hygroscopic growth factor for a aerosol
particle can be approximately expressed as the following (Brock et al., 2016): $g(\text{RH}) \cong (1 +$
$\kappa\frac{RH}{100-RH})^{1/3}$. Enhancement factor in volume can be expressed as the cube of g(RH). Of
particular note is that aerosol particles larger than 100 nm contribute the most to $\sigma_{sp}$ and
$V_a(\text{dry})$, which means that if κ values of aerosol particles of different sizes are the same, then
Vg(RH) can be approximately expressed as $Vg(\text{RH}) = 1 + \kappa\frac{RH}{100-RH}$. In addition, $\sigma_{sp}$ is usually
proportional to $V_a(\text{dry})$ which indicates that the relative change in $\sigma_{sp}$ due to aerosol water
uptake is roughly proportional to relative change in aerosol volume. Therefore, $f(\text{RH})$ might
also be well described by using the formula form of equation (2). Previous studies have shown
that this parameterization scheme can describe $f(\text{RH})$ well (Brock et al., 2016;Kuang et al.,
2017a).
During processes of measuring $f(\text{RH})$, the sample RH in the "dry" nephelometer ($RH_0$) is
not zero. According to equation (2), the measured $f(\text{RH})_{measure} = \frac{f(\text{RH})}{f(RH_0)}$ should be fitted using
the following formula:
$$f(\text{RH})_{measure} = (1 + \kappa_{sca}\frac{RH}{100-RH})/(1 + \kappa_{sca}\frac{RH_0}{100-RH_0}) \quad (3)$$
Based on this equation, $\kappa_{sca}$ can be calculated from measured $f(\text{RH})$ directly.





167 The typical value of $RH_0$ measured in the "dry" nephelometer during Wangdu campaign is

168 about 20%. The importance of the $RH_0$ correction changes under different aerosol hygroscopicity

169 and $RH_0$ conditions. The parameter $\kappa_{sca}$ is fitted with and without consideration of $RH_0$ for

170 $f(\mathrm{RH})$ measurements during Wangdu campaign, and the results are shown in Fig.1. The results

171 demonstrate that, overall, the $\kappa_{sca}$ will be underestimated if the influence of $RH_0$ is not

172 considered, and the larger the $\kappa_{sca}$ the more that the $\kappa_{sca}$ will be underestimated.

173 In addition, based on discussions about the physical understanding of equation (2), the

174 Vg(RH) should be well described by the following equation:

$$\mathrm{Vg(RH)} = 1 + \kappa_{Vf}\frac{RH}{100-RH}\quad(4)$$

176 where $\kappa_{Vf}$ is the parameter which fits Vg(RH) best.

177 **3. Results and discussions**

178 **3.1 Estimation of $V_a(\mathrm{dry})$ from measurements of the "dry" nephelometer**

179 The first step of the proposed method is estimating $V_a(\mathrm{dry})$ from measurements of the "dry"

180 nephelometer. The investigation about the relationship between $V_a(\mathrm{dry})$ and parameters

181 measured by the "dry" nephelometer is required. Results of previous studies demonstrated that

182 $\sigma_{sp}$ of aerosol particles is roughly proportional to $V_a(\mathrm{dry})$ (Pinnick et al., 1980). To confirm this

183 conclusion, datasets of concurrently measured $\sigma_{sp}$ (not corrected for angular truncation error)

184 and PNSD of aerosol particles in dry state from D1 are used to investigate the relationships

185 between measured $\sigma_{sp}$ and $V_a(\mathrm{dry})$. The measured $V_a(\mathrm{dry})$ is integrated from simultaneously

186 measured PNSD. To gain a first glimpse about the roughly proportional relationship between $\sigma_{sp}$





and $V_a(\text{dry})$. All valid data points of measured $\sigma_{sp}$ at 550 nm and $V_a(\text{dry})$ are plotted against
each other and presented in Fig.2a. The results demonstrate that the $\sigma_{sp}$ is highly correlated with
$V_a(\text{dry})$, and the square of correlation coefficient between them is 0.92. The roughly
proportional relationship exists between $V_a(\text{dry})$ and $\sigma_{sp}(550\ nm)$. However, the ratio
$\sigma_{sp}(550\ nm)/V_a(\text{dry})$ (hereinafter referred to as $R_{Vsp}$) varies significantly. The $R_{Vsp}$ for points
in Fig.2a range 1.54 to 6.9 $cm^3/(\mu m^3 \cdot Mm)$, and the average $R_{Vsp}$ is 4.35 $cm^3/(\mu m^3 \cdot Mm)$. If
this average $R_{Vsp}$ is used for estimations of $V_a(\text{dry})$ based on measured $\sigma_{sp}(550\ nm)$, large bias
may occur. Datasets of PNSD and $\sigma_{sp}$ measured by the "dry" nephelometer during Wangdu
campaign are used for investigating the performance of using the average $R_{Vsp}$ in Fig.2a for
estimating $V_a(\text{dry})$, and the results are shown in Fig.2b. The x-axis represents measured $V_a(\text{dry})$
which is integrated from measured PNSD. The y-axis represents estimated $V_a(\text{dry})$ with an
average $R_{Vsp}$. The results demonstrate that although a good correlation exists between estimated
$V_a(\text{dry})$ and measured $V_a(\text{dry})$ (square of correlation coefficient between them is 0.83), large
errors might occur, about 30% of data points have relative differences larger than 30%. More
sophisticated method which can partially account for the variation of $R_{Vsp}$ is needed for
estimating $V_a(\text{dry})$ based on measurements of the "dry" nephelometer.
For developing a method which can partially consider the variation of $R_{Vsp}$, factors which
determine the variation in $R_{Vsp}$ should be aware of.  Here, the quantitative relationship between
$V_a(\text{dry})$ and $\sigma_{sp}$ is analyzed. The $\sigma_{sp}$ and $V_a(\text{dry})$ can be expressed as the following:
$$\sigma_{sp} = \int \pi r^2 Q_{sca}(m,r)\text{n(r)dr} \quad (5)$$
$$V_a(\text{dry}) = \int \frac{4}{3}\pi r^3 \text{n(r)dr} \qquad (6)$$



where $Q_{sca}(m,r)$ is scattering efficiency for a particle with refractive index m and particle
radius r, n(r) is the aerosol size distribution. As presented in equation (5) and (6), relating
$V_a$(dry) with $\sigma_{sp}$ involves complex relation between $Q_{sca}(m,r)$ and particle diameter, and this
relationship can be simulated using Mie theory. In consideration of aerosol refractive index at
visible spectral range, aerosol chemical components can be classified into two categories: the
light absorbing component and the almost light non-absorbing components (inorganic salts and
acids, and most of the organic compounds). Near the visible spectral range, the light absorbing
component can be referred to as BC. BC particles are either externally or internally mixed with
other aerosol components. In view of this, $Q_{sca}$ at 550 nm as a function of particle diameter for
four types of aerosol particles is simulated using Mie theory: almost non-absorbing aerosol
particle, BC particle, BC particle core-shell mixed with non-absorbing components and the
radius of inner BC core are 25 nm and 100 nm, respectively. Same with those introduced in
Sect.2.2, used refractive indices of BC and light non-absorbing components are $1.80 - 0.54i$ and
$1.53 - 10^{-7}i$ , respectively. The simulated results are shown in Fig.3a. Near the visible spectral
range, most of ambient aerosol particles are almost non-absorbing, and their $Q_{sca}$ varies more
like the blue line shown in Fig.3a. In the case of the blue line, aerosol particles with diameter less
than about 800 nm, their $Q_{sca}$ increases almost monotonously with the particle diameter and can
be approximately as a linear function to some extent. Fig.3b shows the simulated size-resolved
accumulative contribution to scattering coefficient at 550 nm for all PNSDs measured during
Wangdu campaign. The results indicate that for continental aerosol particles without influences
of dust, in most cases, all particles with diameter less than about 800 nm contribute more than
80% to total $\sigma_{sp}$. Therefore, for equation (5), If we express $Q_{sca}(m,r)$ as $Q_{sca}(m,r) = k \cdot r$,
then equation (5) can be expressed as the following:



$$\sigma_{sp} = \text{k} \cdot \int \pi r^3 \text{n(r)dr} \qquad (7)$$
This explains why $\sigma_{sp}(550\ nm)$ is roughly proportional to $V_a(\text{dry})$. However, the value k varies
a lot for different particle diameters, which lead to the $R_{Vsp}$ affected greatly by the PNSD which
determines weights of influences of aerosol particles with different diameters on $R_{Vsp}$. The
difference between the blue line and black line shown in Fig.3a indicates that fraction of
externally mixed BC particles in all particles and their sizes will impact on $R_{Vsp}$ largely. The
difference between the black line and the red line as well as the difference between the solid red
line and the dashed red line shown in Fig.3a indicate that how BC mixed with and how much BC
core-shell mixed with other components also exert significant influences on $R_{Vsp}$. In summary,
the variation of $R_{Vsp}$ is mainly determined by variations in PNSD, mass size distribution and
mixing state of BC. It is difficult to find a simple functional relationship between measured $\sigma_{sp}$
and $V_a(\text{dry})$.

The "dry" nephelometer provides not only one single $\sigma_{sp}$ at 550 nm, it measures six

parameters including $\sigma_{sp}$ and back scattering coefficients ($\sigma_{bsp}$) at three wavelengths. The
Ångström exponent calculated from spectral dependence of $\sigma_{sp}$ provide information on mean
predominant aerosol size and is associated mostly with PNSD. However, the mass size
distribution and mixing state of BC also impact on Ångström exponent. The variation of the
hemispheric backscattering fraction (HBF) which is the ratio between $\sigma_{bsp}$ and $\sigma_{sp}$, is essentially
related with mass size distribution and mixing state of BC if the PNSD is fixed (Ma et al., 2012).
If the PNSD and mass size distribution of BC are fixed, higher HBF at 550 nm means that BC
particles are more internally (core-shell) mixed with other aerosol components (Ma et al., 2012).
Hence, variations in both Ångström exponent and HBF are associated with PNSD, mass size





distribution and mixing state of BC. As a result, the Ångström exponent and HBF together
might constrain the variation of $R_{Vsp}$ better. In keeping with this philosophy, $R_{Vsp}$ shown in
Fig.2a are spread into a two-dimensional gridded plot as shown in Fig.4a. Ångström exponent
values are calculated based on concurrently measured $\sigma_{sp}$ at 450 nm and 550 nm from TSI 3563
nephelometer. In Fig.4a, two regions are distinctly differed. In general, when HBF at 550 nm is
larger than 0.14 and   Ångström exponent is larger than 1, the $R_{Vsp}$ tends to be much smaller.
This can be qualitatively understood. For the case of the blue line shown in Fig.3a, if particle
diameter is smaller than about 750 nm, overall, the k value is larger if the particle diameter is
larger. Smaller Ångström exponent and HBF at 550 nm together correspond to relatively larger
particle diameter and therefore larger $R_{Vsp}$. However, more details about the average variation
pattern of $R_{Vsp}$ with changes of HBF at 550 nm and Ångström exponent are difficult to be
disentangled, due to that influences of PNSD, mass size distribution and mixing state of BC on
$R_{Vsp}$ are highly nonlinear. The percentile value of standard deviation of $R_{Vsp}$ values within each
grid of Fig.4a divided by their average is shown in Fig.4b. If HBF at 550 nm is less than 0.13, in
most cases, percentile values shown in Fig.4b are less than 7%, which means that in this region
$R_{Vsp}$ varies little within each grid. However, if HBF at 550 nm is larger than 0.14, in most cases,
percentile values shown in Fig.4b are near or even larger than 20%, which means that in this
region even HBF and Ångström exponent are fixed, $R_{Vsp}$ still varies a lot. These results imply
that if using results shown in Fig.4a as a look up table for estimating $R_{Vsp}$, large bias may occur
when HBF at 550 nm is larger than 0.14.

Datasets of $\sigma_{sp}$ and $\sigma_{bsp}$ measured by the "dry" nephelometer and concurrently measured

PNSD during Wangdu campaign are used for verifying the performance of using results shown



in Fig.4a as a look up for estimating $R_{Vsp}$ and further estimating $V_a(\mathrm{dry})$, and results are shown
in Fig.5a. Compared with the results shown in Fig.2b, the look up table method has improved the
estimation of $V_a(\mathrm{dry})$ markedly (square of correlation coefficient between measured and
estimated $V_a(\mathrm{dry})$ increased from 0.83 to 0.9). It is noticeable that for points with HBF at 550
nm larger than about 0.13, $V_a(\mathrm{dry})$ are systematically underestimated. This result is consistent
with the previous analysis that if using results shown in Fig.4a as a look up table for estimating
$R_{Vsp}$, large bias may occur when HBF at 550 nm is larger than 0.14.

Six parameters are measured by the "dry" nephelometer, however, only three parameters

including $\sigma_{sp}$ at 450 nm and 550 nm, and $\sigma_{bsp}$ at 550 nm are used if using the look up table
shown in Fig.4a for estimating $V_a(\mathrm{dry})$. It can be seen from the results shown in Fig.4b, when
the HBF at 550 nm is larger than 0.14, variations in $R_{Vsp}$ are poorly constrained. Based on the
improvement achieved by using a look up table, we speculate that if all six parameters measured
by the "dry" nephelometer are used together, then HBF at three wavelengths and Ångström
exponent calculated both from $\sigma_{sp}$ at 450 nm and 550 nm and $\sigma_{sp}$ at 550 nm and 700 nm
together can constrain variation in $R_{Vsp}$ better. Machine learning methods which can handle
many input parameters are capable of learning from historical datasets and then make predictions
are powerful tools for tackling highly nonlinear problems. In the light of this, the idea came out
that predicting $V_a(\mathrm{dry})$ based on six optical parameters measured by the "dry" nephelometer
might be accomplished by using a machine learning method. In this paper, we choose the
machine learning function RidgeCV (ridge regression) from the linear model of module Scikit-
learn of computer language Python (Pedregosa et al., 2011) for training the historical datasets of
concurrently measured $V_a(\mathrm{dry})$ and six raw parameters measured by the "dry" nephelometer

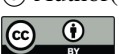



from several field campaigns (Corresponding to data points shown in Fig.2a). Measurements
during Wangdu campaign again are used for evaluating this machine learning method and the
results are shown in Fig.5b. Compared with results shown in Fig.5a, the estimation of $V_a$(dry) is
further improved, not only reflected in the increase of square of correlation coefficient, but also
reflected in the change of the slope. And almost all points with HBF at 550 nm larger than 0.13
distributed within or near the 20% relative difference line. For the machine learning method, the
square of correlation coefficient between measured and estimated $V_a$(dry) is 0.93, with 75% and
43% points have absolute relative differences less than 20% and 10%, respectively. And the
standard deviations of absolute and relative differences between measured and estimated $V_a$(dry)
are 8.4 $\mu m^3/cm^3$ and 10%, respectively.

Measured PNSDs and values of $\sigma_{sp}$ at 550 nm during Wangdu campaign are shown in

Fig.6a and Fig.6b, respectively. The results show that new particle formation phenomena are
frequently observed during Wangdu campaign. In addition, both time series of estimated values
of $V_a$(dry) using the machine learning method and time series of $V_a$(dry) which are integrated
from measured PNSDs are shown in Fig.6c. The results demonstrate that overall, under different
pollution levels and during periods with and without new particle formation phenomena,
predicted $V_a$(dry) agrees well with measured $V_a$(dry). If a reasonable aerosol density is
assumed, measurements from a three-wavelength nephelometer can also be used to estimate total
mass concentrations of ambient aerosol particles in dry state.

Machine learning methods do not explicitly express relationships between many variables,

however, they learn and implicitly construct complex relationships among variables from
historical datasets. Many different and comprehensive machine learning methods are developed



for diverse applications, and can be directly used as a tool for solving a lot of nonlinear problems
which may not be mathematically well understood. We suggest that using machine learning
method for estimating $V_a$(dry) based on measurements of the "dry" nephelometer. The way of
estimating $V_a$(dry) with machine learning method might be applicable for different regions
around the world if used estimators are trained with corresponding regional historical datasets.
**3.2 Bridge the gap between $f$(RH) and Vg(RH)**

The approximate proportional relationship between $\sigma_{sp}$ and $V_a$(dry) introduced in Sect.3.1

is only applicable for aerosol particles of constant refractive index, which is not the case for
aerosol particles growing by addition of water with increasing RH (Hegg et al., 1993). As aerosol
particles grow under conditions of increasing RH, the aerosol scattering efficiency change
nonlinearly and can even decrease. It is difficult to use the same method as introduced in Sect.3.1
to estimate the total aerosol volume of aerosol particles in ambient RH conditions. If Vg(RH) can
be directly estimated from measured $f$(RH), then the ALWC can be estimated. Relating $f$(RH)
to Vg(RH) involves complicated variations of aerosol scattering efficiency as a function of
growing particle diameter due to aerosol water uptake, and this relationship can be described
using Mie theory and κ-Köhler theory.  As discussed in Sect.2.4, $f$(RH) and Vg(RH) can be
described by the formula form of equation (2) and (4). To consolidate this conclusion, a
simulative experiment is conducted. In the simulative experiment, average PNSD in dry state and
mass concentration of BC during the Haze in China (HaChi) campaign (Kuang et al., 2015) are
used. During HaChi campaign, size-resolved $\kappa$ distributions are derived from measured size-
segregated chemical compositions (Liu et al., 2014) and their average is used in this experiment
to account the size dependence of aerosol hygroscopicity. Modelled results of $f$(RH) and



Vg(RH) are shown in Fig.7. Results demonstrate that modelled $f$(RH) and Vg(RH) can be well
parameterized using the formula form of equation (2) and (4). Fitted values of $\kappa_{sca}$ and $\kappa_{Vf}$ are
0.227 and 0.285, respectively. This result indicates that if linkage between $\kappa_{sca}$ and $\kappa_{Vf}$ is
established, measurements of $f$(RH) can be directly related to Vg(RH).
Many factors have significant influences on the relationships between $f$(RH) and Vg(RH),
such as PNSD, BC mixing state and the size-resolved aerosol hygroscopicity. To gain insights
into the relationships between $\kappa_{sca}$ and $\kappa_{Vf}$, a simulative experiment using Mie theory and κ-
Köhler theory is designed. In this experiment, all PNSDs at dry state along with mass
concentrations of BC from D2 are used, characteristics of these PNSDs can be found in Kuang et
al. (2017a). As to size-resolved aerosol hygroscopicity, a number of size-resolved $\kappa$ distributions
were derived from measured size-segregated chemical compositions during HaChi campaign
(Liu et al., 2014). Their results demonstrate that overall, size-resolved $\kappa$ distributions have three
modes: highly hygroscopic mode with diameters of aerosol particles ranging from 150 nm to 1
μm, less hygroscopic mode with diameters of aerosol particles less than 150 nm and nearly
hydrophobic mode with diameters of aerosol particles larger than 1 μm. The shape of the average
size-resolved $\kappa$ distribution during HaChi campaign (black line shown in Fig.9a) is used in the
designed experiment. Other than the shape of size-resolved $\kappa$ distribution, the overall aerosol
hygroscopicity which determines the magnitude of $f$(RH) also have large impacts on the
relationship between $\kappa_{sca}$ and $\kappa_{Vf}$. In view of this, ratios range from 0.05 to 2 with an interval of
0.05 are multiplied with the aforementioned average size-resolved $\kappa$ distribution (the black line
shown in Fig.9a) to produce a number of size-resolved $\kappa$ distributions which represent aerosol
particles from nearly hydrophobic to highly hygroscopic. During simulating processes, each





PNSD is modelled with all produced size-resolved $\kappa$ distributions. In the following, the ratio
$\kappa_{Vf}/\kappa_{sca}$ termed as $R_{Vf}$ is used to indicate the relationship between $\kappa_{sca}$ and $\kappa_{Vf}$.
In consideration of that values of Ångström exponent contain information about PNSD
(Kuang et al., 2017a) and values of $\kappa_{sca}$ represent overall hygroscopicity of ambient aerosol
particles, and both the two parameters can be directly calculated from measurements of a three-
wavelength humidified nephelometer system (Kuang et al., 2017a). Simulated $R_{Vf}$ values are
spread into a two-dimensional gridded plot. The first dimension is Ångström exponent with an
interval of 0.02 and the second dimension is $\kappa_{sca}$ with an interval of 0.01. Average $R_{Vf}$ value
within each grid is represented by color and shown in Fig.8a. Values of Ångström exponent
corresponding to used PNSDs are calculated from simultaneously measured $\sigma_{sp}$ values at 450
nm and 550 nm from TSI 3563 nephelometer. Results shown in Fig.8a exhibit that both PNSD
and overall aerosol hygroscopicty have significant influences on $R_{Vf}$. Simulated values of $R_{Vf}$
range from 0.8 to 1.7 with an average of 1.2. Overall, $R_{Vf}$ value is lower when value of
Ångström exponent is larger. With respect to influences of $\kappa_{sca}$ on $R_{Vf}$, if Ångström exponent
is larger than about 1.1, $\kappa_{sca}$ have small influences on $R_{Vf}$ while its influence is remarkable
when Ångström exponent is lower than 1.1. In addition, the percentile value of standard
deviation of $R_{Vf}$ values within each grid divided by its average is shown in Fig.8b. In most
cases, these percentile values are less than 10% (about 90%) which demonstrates that $R_{Vf}$ varies
little within each grid shown in Fig.8a. This implies that results of Fig.8a can serve as a look up
table to estimate $R_{Vf}$ and thereby $\kappa_{Vf}$ values can be directly predicted from measurements of a
three-wavelength humidified nephelometer system.



For the look up table shown in Fig.8a, a fixed size-resolved $\kappa$ distribution is used, which
might not be able to capture variations of $R_{Vf}$ induced by different types of size-resolved $\kappa$
distributions under different PNSD conditions. A simulative experiment is conducted to
investigate the performance of this look up table. In this experiment, the following datasets are
used: PNSDs and mass concentrations of BC from D2 (the number of used PNSD is 11996), and
size-resolved $\kappa$ distributions from HaChi campaign (Liu et al., 2014) which are presented in
Fig.9a (the number is 23). Results shown in Fig.9a imply that the shape of size-resolved $\kappa$
distribution has no apparent correlation with pollution degrees and varies a lot. During the
simulating processes, for each PNSD, it is used to simulate $R_{Vf}$ values corresponding to all used
size-resolved $\kappa$ distributions, therefore, 275908 $R_{Vf}$ values are modelled. Also, modelled values
of $\kappa_{sca}$ and corresponding values of modelled Ångström exponent are together used to estimate
$R_{Vf}$ values using the look up table shown in Fig.8a. Results of relative differences between
estimated and modelled $R_{Vf}$ values under different pollution conditions are shown in Fig.9b.
Overall, 88% of points have absolute relative differences less than 15%, and 68% of points have
absolute relative differences less than 10%. This look up table performs better when the air is
relatively polluted.
**3.3 Estimation of the ambient ALWC**
During the Wangdu campaign, there are ten days of measurements from the humidified
nephelometer system are available. Values of $\kappa_{sca}$ are first fitted from observed $f(RH)$ curves
and then linearly interpolated to times of ambient RH points (one $f(RH)$ curve lasts about 45
minutes, the time resolution of used ambient RH is five minutes), and the results are shown in
Fig.20a. The RH range of one $f(RH)$ cycle is about 50% to 90%. The estimated values of $\kappa_{Vf}$



using results shown in Fig.20a as a look up table is also shown in Fig.20a. During this
observation period, $\kappa_{sca}$ ranges from 0.05 to 0.3 with an average of 0.19. Estimated values of
$R_{Vf}$ ranges from 0.86 to 1.47, with an average of 1.15. Estimated values of $\kappa_{Vf}$ ranges from 0.05
to 0.35, with an average of 0.22. Time series of ambient RH is shown in Fig.20b, and RH points
with RH larger than 95% are excluded because the measurements of ambient RH at this range is
highly uncertain. With estimated values of $\kappa_{Vf}$ and measured ambient RH, Vg(RH) of aerosol
particles in ambient RH states can be estimated. Then, with measured $V_a$(dry) (shown in
Fig.20c) which is integrated from measured PNSD, water volumes of ambient aerosol particles
are estimated and shown in Fig.20c. During this observation period, estimated water volume of
ambient aerosol particles mainly range from 1 to 300 $\mu m^3/cm^3$, with an average of 42
$\mu m^3/cm^3$.
**3.4 Uncertainty analysis**

According to the equation $Vg(RH) = 1 + \kappa_{Vf}\frac{RH}{100-RH}$, the estimated volume of aerosol

liquid water ($V_{water}$) can be expressed as: $V_{water} = V_a(dry) \cdot \kappa_{sca} \cdot R_{Vf} \cdot \frac{RH}{100-RH}$. Neglecting
measurement uncertainty of ambient RH, uncertainties contribute to $V_{water}$ include uncertainty
of $V_a$(dry), uncertainties of $\kappa_{sca}$ and $R_{Vf}$.

Results introduced in Sect.3.1 suggest that using the machine learning method to predict

$V_a$(dry) from measurements of a three-wavelength nephelometer is feasible but non-negligible
bias still exists between measured and estimated $V_a$(dry). The standard deviation of relative
differences between measured and estimated $V_a$(dry) is 15%. If using triple the standard
deviation (99% of points locate within this range) as the uncertainty of this method, the

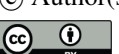



uncertainty is 45%. Here, sources of this large bias is discussed. The $V_a$(dry) is determined from
PNSD which is high-dimensional. Six parameters provided by the "dry" nephelometer cannot
accurately constrain $R_{Vsp}$ in the machine learning method. This should be the largest uncertainty
source. In addition, used datasets for training the estimator carried some uncertainties which
should also influence the performance of the estimator. Using a Monte Carlo method based on
uncertainties of measured PNSD (see Table 3 of Ma et al. (2014) for the uncertainty parameters
of PNSD), $V_a$(dry) integrated from measured PNSD have uncertainty of about 5%. The TSI
3563 nephelometer also carry some uncertainties, it is about 9% (Heintzenberg et al.,
2006;Sherman et al., 2015). Their uncertainties will propagate in the processes of training and
verifying the estimator. If the datasets for training the estimator are more comprehensive (like a
year of observation in several sites), the uncertainty of this machine learning method might be
smaller.

The $\kappa_{sca}$ is directly fitted from $f$(RH) measurements. Results of Titos et al. (2016)

demonstrate that, for moderately hygroscopic aerosols (e.g., $f$(RH = 80%) less than 2.2),
$f$(RH) errors are around 15%. Since most values of $f$(RH = 80%) observed on continental
regions are less than 2.2 (Zhang et al., 2015;Titos et al., 2016), 15% is used as the uncertainty of
$f$(RH) as well as $\kappa_{sca}$.

As to uncertainty of estimated $R_{Vf}$. Many factors exert influences on $R_{Vf}$, such as PNSD,

mixing state of BC and size-resolved $\kappa$ distribution. If using the 99% line (triple the standard
deviation) shown in Fig.9b as uncertainties of $R_{Vf}$ from influences of size-resolved $\kappa$ distribution
and PNSD, then this aspect of uncertainties of $R_{Vf}$ under different pollution conditions range
from 17% to 49%. Additionally, the mixing state of BC can also impact on $R_{Vf}$. In this study, the



BC is assumed to be half externally and half coreshell mixed with other aerosol components. A
simple simulative test is performed to investigate the influence of BC mixing state on $R_{Vf}$. In
this test, we simulated $R_{Vf}$ values for three kinds of BC mixing states: external; half external and
half coreshell; core-shell, and the average PNSD and mass concentration of BC during Wangdu
campaign are used. Simulated $R_{Vf}$ values for these three mixing state are 1.13, 1.18 and 1.25,
respectively. Thus, we consider 6% as the uncertainty of $R_{Vf}$ caused by the variation of BC
mixing state. The synthesized uncertainties of estimated $R_{Vf}$ under different pollution levels are
presented in Fig.21, which have considered the variations of BC mixing state and size-resolved $\kappa$
distribution and PNSD. Uncertainties of estimated $R_{Vf}$ by using the look up table shown in
Fig.8a range from 18% to 49.4%.
With estimated uncertainties of $V_a(\text{dry})$, $\kappa_{sca}$ and $R_{Vf}$, the uncertainties of estimated $V_{water}$
under different pollution levels can be estimated. In the processes of estimating $V_{water}$, two
methods can be used to estimate $V_a(\text{dry})$. The first mehtod is estimating $V_a(\text{dry})$ from
measurements of the three-wavelength "dry" nephelometer (Method 1). However, if PNSD is
available, $V_a(\text{dry})$ can be directly integrated from measured PNSD (Method 2). The calculated
uncertainties of $V_{water}$ under different pollution levels with $V_a(\text{dry})$ estimated from these two
methods are presented in Fig.21. For Method 1, uncertainties of estimated $V_{water}$ range from
24% to 52%, with an average of 31%. For Method 2, uncertainties of estimated $V_{water}$ range
from 51% to 68%%, with an average of 55%. Compared to clean conditions, the uncertainty of
estimated $V_{water}$ is smaller when the air is highly polluted. We recommend that if measured
PNSD is available, $V_a(\text{dry})$ should be calculated from measured PNSD, otherwise, $V_a(\text{dry})$ can
be estimated from measurements of the "dry" nephelometer.

The method proposed in this research is based on datasets of PNSD, $\sigma_{sp}$ and size-resolved $\kappa$ distribution which are measured on the NCP without influences of dust and sea salt. Cautions should be exercised if using the proposed method to estimate the ALWC when the air mass is influenced by sea salt or dust. The way of estimating $V_a(\text{dry})$ with machine learning method might be applicable for different regions around the world. However, the used estimator from machine learning should be trained with corresponding regional historical datasets. The way of connecting $f(\text{RH})$ to $\text{Vg}(\text{RH})$ might also be applicable for other continental regions. Still, we suggest that the used look up table is simulated from regional historical datasets.

Note that the humidified nephelometer usually operates with RH less than 95%. Aerosol water, however, increase dramatically with increasing RH when RH is greater than 95%. Such high RH conditions can occur during the haze events. This may limits the usage of the proposed method when ambient RH is extremely high. As discussed in Sect.2.4, the proposed way of connecting $f(\text{RH})$ and $\text{Vg}(\text{RH})$ is based on the κ-K$\ddot{o}$hler theory. If κ does not change with RH, the proposed method should be applicable when RH is higher than 95%, even the measurements of humidified nephelometer system are conducted when RH is less than 95%. Many studies have done researches about the change of κ with the changing RH (Rastak et al., 2017;Renbaum-Wolff et al., 2016), their results demonstrate that the κ changes with increasing RH. However, few studies have investigated the variation of κ of ambient aerosol particles with changing RH when RH is less than 100%. Liu et al. (2011) have measured κ of ambient aerosol particles at different RHs (90%, 95%, 98.5%) on the NCP. Their results demonstrated that κ at different RHs differ little for ambient aerosol particles with different diameters. Results of Kuang et al. (2017b) indicated that κ values retrieved from $f(\text{RH})$ measurements agree well with κ values at RH of 98% of aerosol particles with diameter of 250 nm. In this respect, the proposed method might be



applicable even when ambient RH is extremely high for ambient aerosol particles on the NCP.
Moreover, for calculating the ambient ALWC, the measured ambient RH is required. However,
if the ambient RH is higher than 95%, the measured ambient RH with current techniques is
highly uncertain. Given this, cautions should be exercised if the ambient ALWC is calculated
when the ambient RH is higher than 95%.

**4. conclusions**

In this paper, a novel method is proposed to calculate ALWC based on measurements of a

three-wavelength humidified nephelometer system. Two critical relationships are required in this
method. One is the relationship between $V_a(\text{dry})$ and measurements of the "dry" nephelometer.
Another one is the relationship between Vg(RH) and $f(\text{RH})$. The ALWC can be calculated from
the estimated $V_a(\text{dry})$ and Vg(RH).

Previous studies have shown that an approximate proportional relationship exists between

$V_a(\text{dry})$ and corresponding $\sigma_{sp}$, especially for fine particles (particle diameter less than 1 μm).
However, PNSD and other factors still have significant influences on this proportional relationship.
It is difficult to directly estimate $V_a(\text{dry})$ from measured $\sigma_{sp}$. In this paper, an estimator from
machine learning procedure is used to estimate $V_a(\text{dry})$ based on measurements of a three-
wavelength nephelometer. This estimator is trained with datasets of PNSD and $\sigma_{sp}$ from several
field campaigns conducted on the NCP. This method is then validated using measurements from
Wangdu campaign. The square of correlation coefficient between measured and estimated $V_a(\text{dry})$
is 0.93.



The relationship between $Vg(RH)$ and $f(RH)$ is then investigated by conducting a simulative
experiment. It is found that the complicated relationship between $Vg(RH)$ and $f(RH)$ can be
disentangled by using a look up table, and parameters required in the look up table can be directly
calculated from measurements of a three-wavelength humidified nephelometer system. Given that
the $V_a(dry)$ can be estimated from a three-wavelength "dry" nephelometer, the ambient ALWC
can be estimated from measurements of a three-wavelength humidified nephelometer system in
conjunction with measured ambient RH. During Wangdu campaign, calculated water volumes of
ambient aerosol particles range from 1 to 300 $\mu m^3/cm^3$, with an average of $42\ \mu m^3/cm^3$.
Results introduced in this research have bridged the gap between $f(RH)$ and $Vg(RH)$. The
advantage of using measurements of a humidified nephelometer system to estimate ALWC is
that this technique has a fast response time and can provide continuous measurements of the
changing ambient conditions. The new method proposed in this research will facilitate the real-
time monitoring of the ambient ALWC and further our understanding of roles of ALWC in
atmospheric chemistry, secondary aerosol formation and climate change.
**Acknowledgments**
This work is supported by the National Natural Science Foundation of China (41590872,
41375134). The data used are listed in the references and a repository at
http://pan.baidu.com/s/1c2Nzc5a.

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







Table 1 Abbreviations

| | |
|---|---|
| RH | relative humidity |
| $f(\text{RH})$ | aerosol light scattering enhancement factor at 550 nm |
| ALWC | aerosol liquid water content |
| $V_a(\text{dry})$ | total volume of ambient aerosol particles in dry state |
| Vg(RH) | aerosol volume enhancement factor due to water uptake |
| NCP | North China Plain |
| HTDMA | humidified tandem differential mobility analyser system |
| PNSD | particle number size distribution |
| BC | black carbon |
| g(RH) | hygroscopic growth factor |
| APS | Aerodynamic Particle Sizer |
| SMPS | scanning mobility particle size spectrometer |
| $\sigma_{sp}$ | aerosol light scattering coefficient |
| $\sigma_{bsp}$ | aerosol back scattering coefficient |
| $\sigma_{ext}$ | aerosol extinction coefficient |
| $R_{Vsp}$ | $\sigma_{sp}(550\,nm)/V_a(\text{dry})$ |
| F1 to F5 | referred as to five field campaigns listed in Table S1 |
| D1 | PNSD, BC and nephelometer measurements from field campaigns F1 to F4 |
| D2 | PNSD, BC and nephelometer measurements from F2, F4 and F5 |









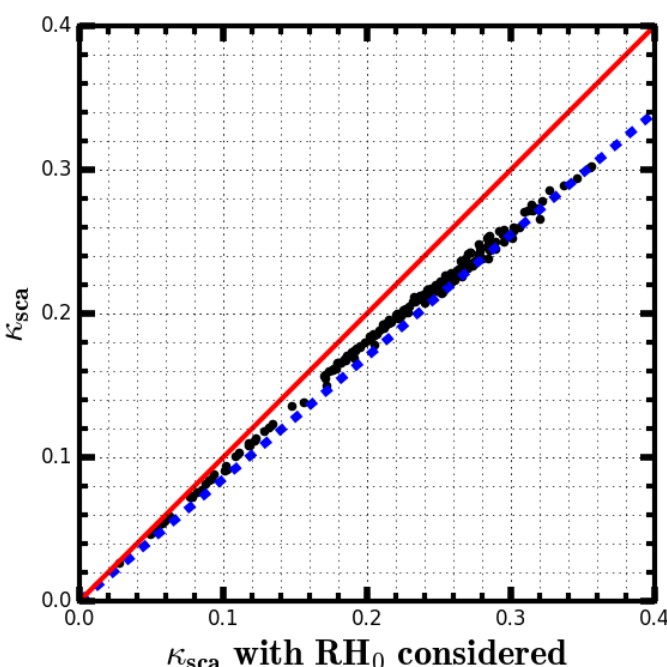


**Figure 1**. X-axis and y-axis represent $\kappa_{sca}$ are fitted with and without consideration of $RH_0$ in the "dry" nephelometer, respectively. The red line is 1:1 line, the blue dashed line is the 15% relative difference line.











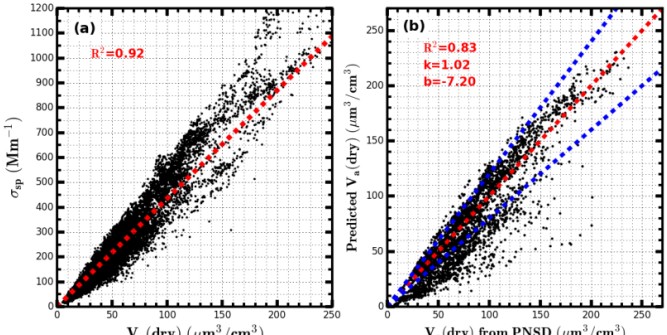


**Figure 2**. (a) Scatter plot of all valid data points of $V_a(\text{dry})$ and $\sigma_{sp}$ at 550 nm from D1, dashed red line is the
line whose slope is equal to the average ratio $\sigma_{sp}(550\,nm)/V_a(\text{dry})(R_{Vsp})$. (b) The comparison between
$V_a(\text{dry})$ estimated from a fixed average $R_{Vsp}$ of (a) and measured $V_a(\text{dry})$. In figure (a), the dashed red line is
the line whose slope is the average ratio $R_{Vsp}$. In figure (b), red line is the 1:1 line, two dashed blue line are lines
with relative difference of 20%. $R^2$ is the suqare of correlation coefficient, k is the slope, b is the intercept.





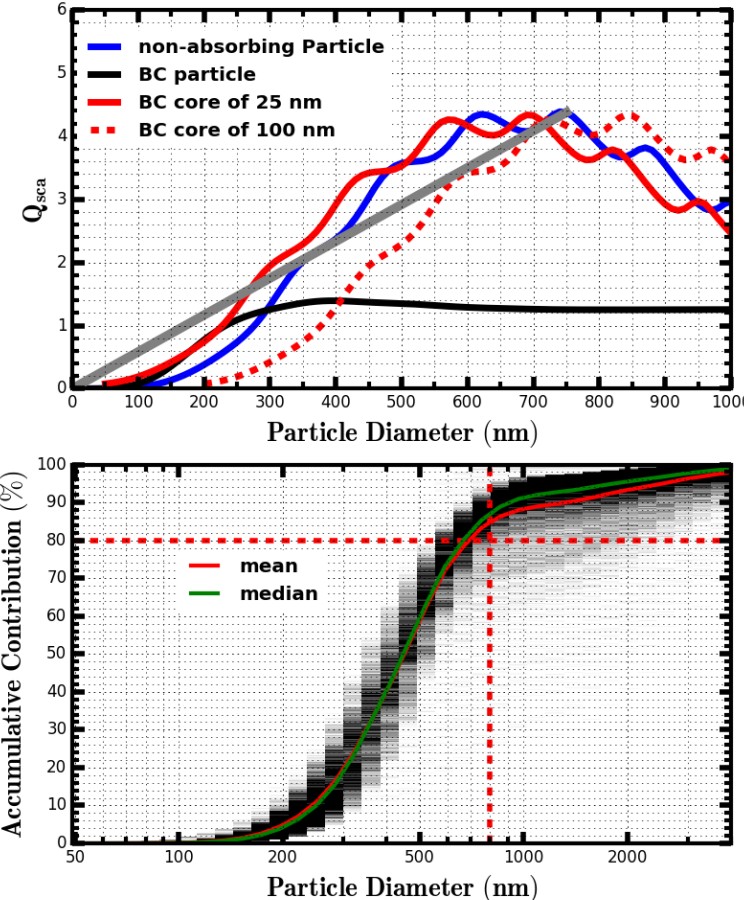


**Figure 3**. (a) $Q_{sca}$ at 550 nm as a function of particle diameter for four types of aerosol particles, the gray line
corresponds to the fitted linear line for the case of non-absorbing particle when particle diameter is less than 750
nm. (b) Simulated size-resolved accumulative contribution to scattering coefficient at 550 nm for all PNSDs
measured during Wangdu campaign, the color scales (from light gray to black) represent occurrences.

710

711





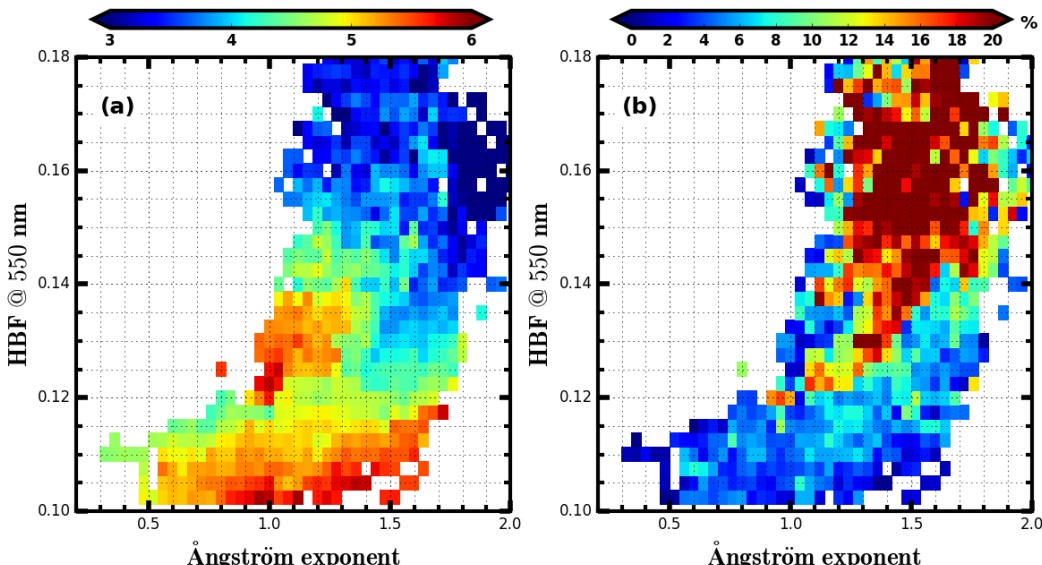

712

**Figure 4**. (a) Colors represent $R_{Vsp}$ values and the color bar is shown on the top of this figure, x-axis
represents Ångström exponent and y-axis represents HBF at 550 nm. (b) Meanings of x-axis and y-axis are
same with them in (a), however, color represents the percentile value of the standard deviation of $R_{Vsp}$ values
within each grid divided by their average.





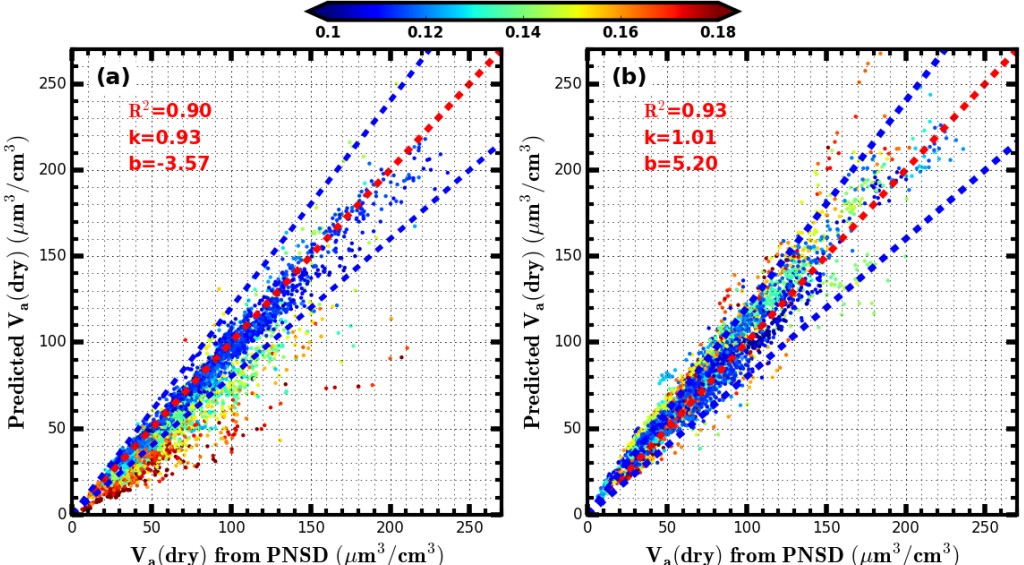

718

**Figure 5**. Comparisons between predicted and measured $V_a(\text{dry})$, the red dashed line is the 1:1 line, two blue
dashed line shown in (a) and (b) are lines with relative difference of 20%. Colors of scattered points in (a) and
(b) represent corresponding values of HBF at 550 nm, and the color bar is shown on the top. (a) $V_a(\text{dry})$ in the
y-axis is predicted by using results shown in Fig.4a as a look up table. (b) $V_a(\text{dry})$ in the y-axis is predicted by
using the machine learning method.




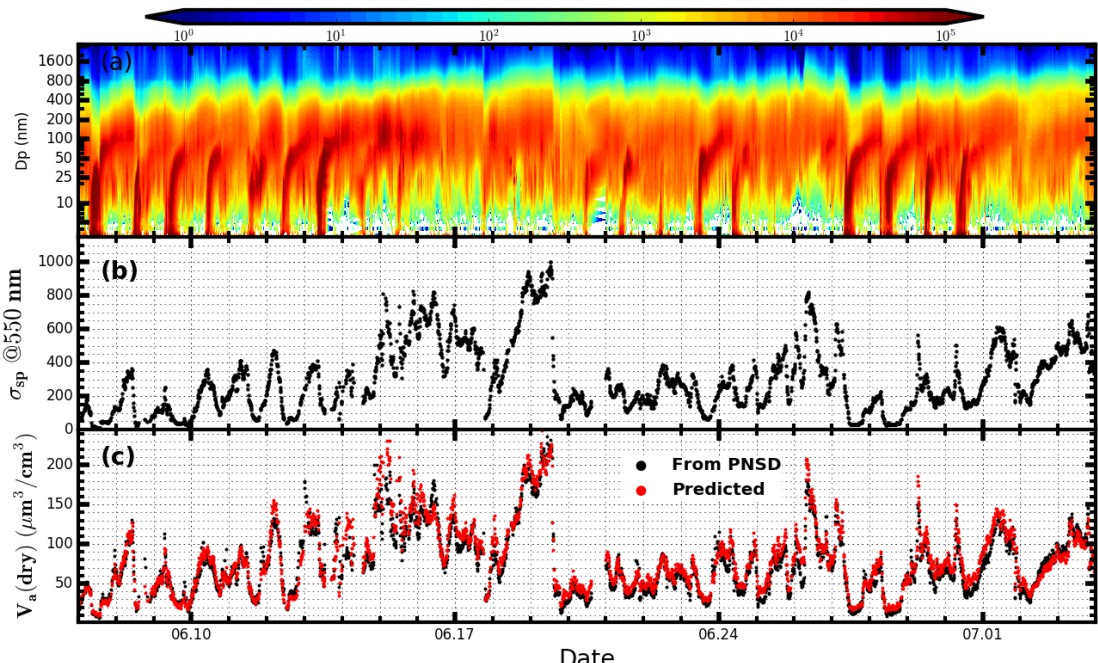


**Figure 6.** Measurements of PNSD and $\sigma_{sp}$ during Wangdu campaign; (a) Time series of PNSD in dry state,

colors represent $dN/dlog(Dp)(cm^{-3})$; (b) Time series of measured $\sigma_{sp}$ at 550 nm; (c) $V_a$(dry) integrated from

measured PNSD and $V_a$(dry) predicted by using the machine learning method.













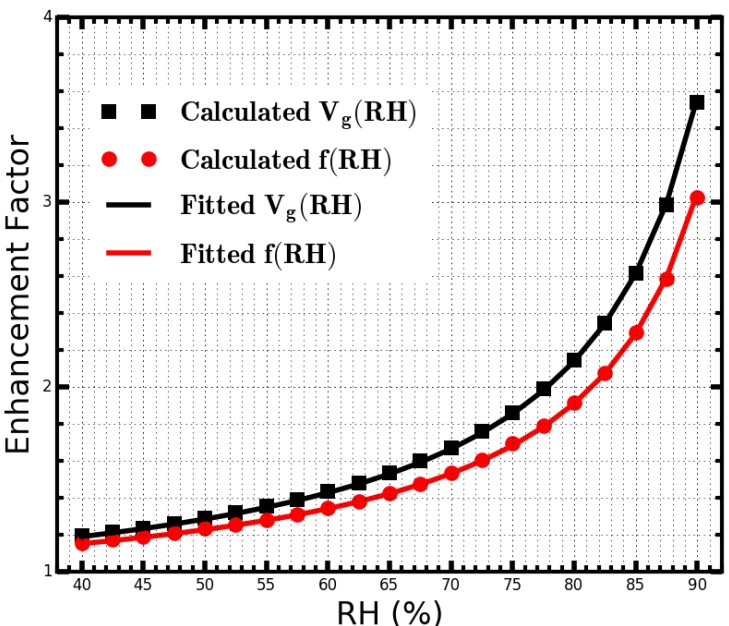


**Figure 7**. Modelled $f$(RH) and Vg(RH) (scatter points) and fitted $f$(RH) and Vg(RH) (solid lines) using

formula form of equation (2).





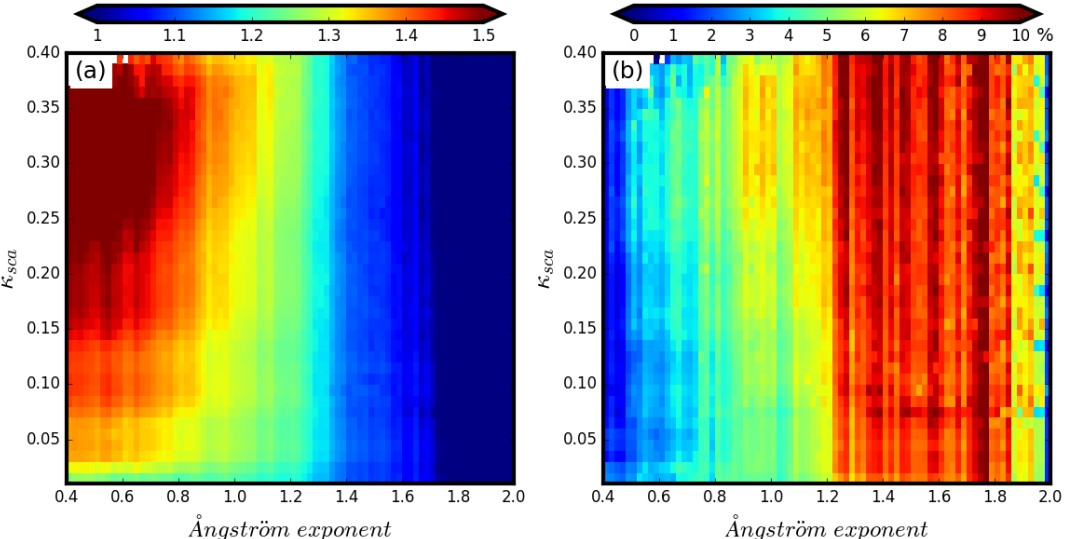


**Figure 8**. (a) Colors represent $R_{Vf}$ values and the colorbar is shown on the top of this figure, x-axis represents

Ångström exponent and y-axis represents $\kappa_{sca}$. (b) Meanings of x-axis and y-axis are same with them in (a),

however, color represents the percentile value of the standard deviation of $R_{Vf}$ values within each grid divided

by their average.


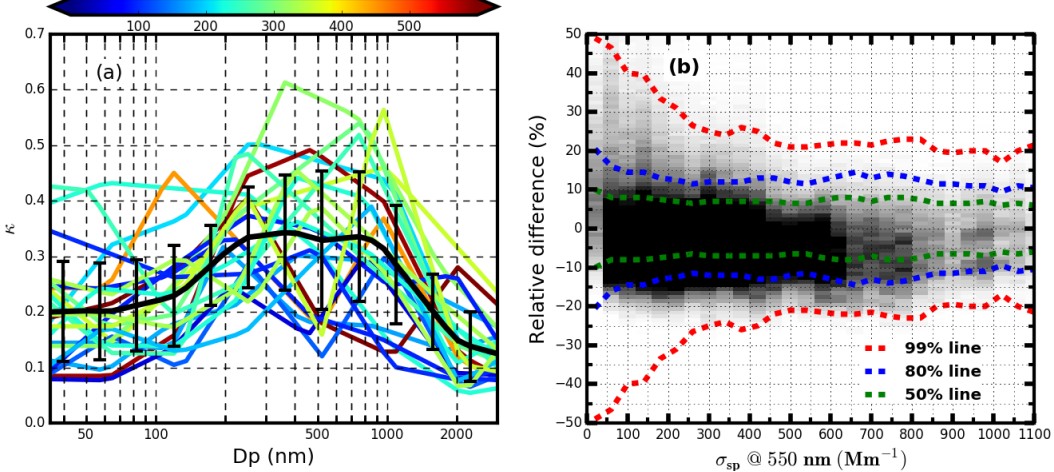


**Figure 9**. (a) All size-resolved $\kappa$ distributions which are derived from measured size-segregated chemical

compositions during HaChi campaign, colors represent corresponding values of average $\sigma_{sp}$ at 550 nm ($Mm^{-1}$),

black solid line is the average size-resolved $\kappa$ distribution and error bars are standard deviations ; (b) The gray





colors represent the distribution of relative differences between modelled and estimated $R_{Vf}$ values, darker grids
have higher frequency, dashed lines with the same color mean that corresponding percentile of points locate
between the two lines.


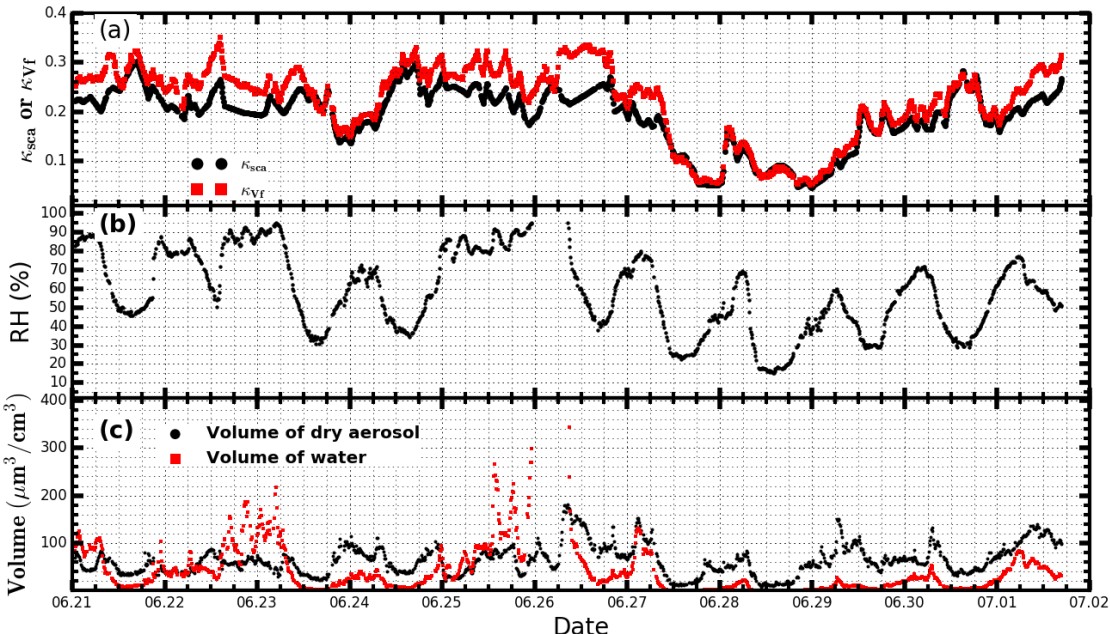


**Figure 10.** (a) Time series of values of $\kappa_{sca}$ fitted from observed $f(\mathrm{RH})$ curves and predicted values of $\kappa_{Vf}$ by
using results shown in Fig.8a as a look up table; (b) Measured ambient RH; (c) Time series of $V_a(\mathrm{dry})$
($\mu m^3/cm^3$) which is integrated from measured PNSD and volume of aerosol liquid water estimated from
combination of $\kappa_{Vf}$ and ambient RH.



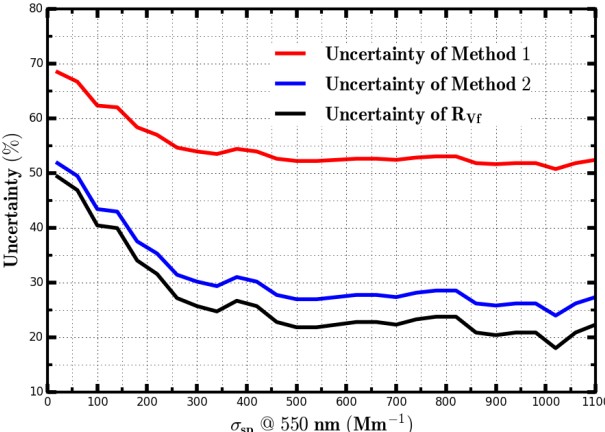

**Figure 11**. Black line corresponds to uncertainty of predicted $R_{Vf}$ by using results shown in Fig.5a as the look
up table. Blue and red lines represent uncertainties of volumes of aerosol liquid water which are estimated from
the following two methods: Method 1 corresponds to $V_a(\text{dry})$ is estimated from the machine learning method,
Method 2 corresponds to $V_a(\text{dry})$ is integrated from the concurrently measured PNSD.







