# Peer review of "A novel method for calculating ambient aerosol liquid water content based on measurements of a humidified nephelometer system"

_Atmospheric Measurement Techniques, 2017_

## Referee Comment (RC1) · Anonymous Referee #2 · 27 Nov 2017

The paper, "A novel method for calculating ambient aerosol liquid water contents based on measurements of a humidified nephelometer system" describes a new technique to measure aerosol liquid water content (LWC). The method described has advantages over traditional methods (ie HTDMA) as it measures LWC of aerosols in real time. In addition, this three wavelength nephelometer system measures the entire size distribution at once without assuming a constant growth factor for the entire distribution, as is the case with previous nephelometer measurements. The LWC of aerosols has important implications for climate forcing and atmospheric chemistry and there is a need for a more accurate measurement to reduce uncertainties.

In general there are large sections of this paper that should be omitted. The content does not support the conclusions of the paper. More importantly, these sections are sometimes confusing and will only distract the reader. This includes the paragraphs describing the relationship between scattering and aerosol volume, which the authors show to be correlated but not easily parameterized. The next section, which describes using the angstrom exponent and HBF to constrain the ratio between scattering and aerosol volume can also be summarized or omitted. The author's conclusion that large bias' will occur if using the "look up table" (fig 4) isn't necessary for the sake of the papers conclusions. More emphasis needs to be placed on what was done that produced usable results.

Finally, the authors describe the machine learning method, which improves the ability to predict the volume of aerosol in the dry state. During the Wangdu campaign (Fig 6), this method is able to reproduce the dry volume very well. My main concern is how applicable this method would be in a different type of environment. How difficult would it be to train the estimator to respond to different data sets and measurement conditions?

The next section describes parametrizing the relationship between f(RH) and humidified aerosol volume using the "look up table" shown in fig 8. I would recommend referring to both fig 8 and fig 4 as something other than a lookup table, which is not an appropriate description for this plot. This approach, once again seems limited by the specific environment. It would be nice if the authors showed results from a different less-polluted region.

The paper has multiple typographical and grammatical errors. Line 365 is a good example of the grammatical/typographical errors that are found throughout. Line 405 references a figure 20a, which doesn't exist. Prior to publication I would recommend careful editing for these errors as well as re-formatting to streamline the paper for only the most pertinent information.

---

## Referee Comment (RC2) · Anonymous Referee #3 · 25 Jan 2018

Date: January 19, 2018 Paper Review: A novel method for calculating ambient aerosol liquid water contents based on measurements of a humidified nephelometer system Authors: Ye Kuang et al.

The authors present both empirical and machine-learning methods for determining an aggregate or effective volume and the aerosol water content from the nephelometer Angstrom and backscattering coefficients. The empirical method uses cross-correlations between the aerosol scattering coefficient, measured volume, backscatter fraction, and Angstrom exponent to estimate the aerosol volume. The machine-learning method uses backscatter and Angstrom exponents to mimic the aerosol scat-

tering to volume ratio. The machine-learning method offers a valuable tool that could be applied to many aspects of atmospheric aerosol and chemical predictions. The paper needs further development. I think it's important to refine the methodology, improve the paper organization, and clarify some of text to make this a stronger paper. The section linking scattering hygroscopic growth to volume hygroscopic growth doesn't follow a valid analysis method. The fRH and gRH data come from different measurement sites. I suggest leaving out the sections on volume hygroscopic growth as well as the discussion on kappa-Kohler. I don't recommend publication until the paper is restructured. I suggest resubmitting the paper after removal of the sections on hygroscopic growth and water content.

The paper needs better organization and clear, step-wise presentation of the methods and results. The methodology is scattered throughout the paper. Description of the results is vague and tends to gloss over important features.

Please edit the English grammar and word order. Avoid run-on sentences. You have a tendency to state conclusions without providing supporting evidence. State the methods used and then the data results. The methods and results are interspersed in the paper, which adds to confusion.

Introduction 1. Although the methodology is different from other inversion techniques, such as those of Ziegar et al., that calculate an "effective" gRH from fRH, the methodology is similar to that used in Aeronet retrievals of the aerosol effective radius from the AOD, Angstrom exponent and asymmetry parameter. There was a paper that attempted to calculate fRH or aerosol water using Aeronet data, but it suffered from low signal and spatial resolution. Can the authors describe how their method is similar to or different from the Aeronet retrievals and also speculate if this method could be used with remote sensing AOD measurements? Below is a link that has links to their "spectral deconvolution algorithm".

https://aeronet.gsfc.nasa.gov/new_web/Documents/Inversion_products_V2.pdf or Atmos. Meas. Tech., 10, 695-708, 2017, https://doi.org/10.5194/amt-10-695-2017

2. The introduction needs to state more about the methodology other than saying it's "a novel method". Add a paragraph describing the two techniques. Describe the empirical model use of size-dependent parameters (backscatter and Angstrom) to predict the ratio of scattering/volume. Describe how machine-learning methods, using large data sets, over a long-time period as input, mimic the system behavior via feed-back loops to predict an output.

3. In Section 2.1 Transfer Table S1 to Section 2.1. You can leave the detailed measurement description in the supplement.

Give general information on the breakdown of the aerosol composition between organics/sulfate/nitrate/dust at this time of year.

Sections 2.2 to 3.2 need to be reorganized. I suggest segregating Section 2 into Section 2.2 is closure, 2.3 is Mie theory, 2.4 Machine learning method

I suggest renaming section 2.2 as "Closure Calculations". Show the scattering closure between the measured and calculated scattering coefficient. The integrity of the volume and fRH closure depends on the scattering closure. Figure 2 would validate the measurements better if 2a showed the scattering closure and 2b showed the scattering vs volume.

Figure 2 shows 2 branches or subsets of the data; one above and a 2nd below the fit line. Is this behavior present in the scattering closure? Do these two branches represent 2 different aerosol types or multiple size modes?

The application of an average Rvsp to estimate the volume doesn't add to the paper and distracts from the other methods. I suggest removing it from the analysis.

Section 2.3 Mie Move equations 5 and 6 to the start of section 2.3 and explain how you relate scattering to volume and the assumptions in this approach. Move the discussion of Mie theory from section 3.1 to section 2.3. State your adaptation of Mie Theory in

a clear, stepwise, logical fashion. Show that going from equations 5 > 7 >6 assumes that Q is roughly linear with r such that Q=k*Q(m).

Describe Mie model using simulated data with 4 aerosol types and results in Figure 3. Describe limitations of assumption that Q is linearly proportional to r.

Describe what the dotted red lines represent in the Figure 3 caption.

Section 2.5 Machine learning

Describing an alternate method of machine learning using size-dependent scattering parameters; Angstrom and backscatter to mimic the measured scattering/volume ratio. Give some background on "machine learning" and the algorithm name. Can you add a simplified algorithm decision tree with basic logic steps or diagram that would help explain the process?

Results and Discussion: Section 3.1 Empirical method Refer back to Figure 2 and need for estimating a variable kscat or R. Introduce the empirical method of determining R from HBF and Angstrom exponent. Explain figure 4.

In your description of the results, use and simple and direct language. Leave out extra information on the impact of BC and mixing state on the HBF and Angstrom exponents until the discussion of the data fit lines as these are secondary contributions.

Explain Figure 5a and variance about fit line in relation to HBF. This variance likely stems from the Angstrom exponent and HBF describing a fraction of the PNSD. Show a plot of HBF(450, 550, 700nm) and Angstrom exponents(450/550, 450/700 and 550/700) versus r for a lognormal size distribution. The plot will show the sensitivities of these parameters to aerosol size.

Section 3.2 Machine learning Explain how machine method uses 6 parameters to describe the aerosol volume relative to the scattering. Explain figures 5b and Figure 6 and how machine learning is an improvement over the empirical method.

[Figure]

**3.3 fRH and Vrh**

The method of simulating the Kd size distribution from variations of the average and then applying this to the measured size distribution to obtain a 4 modeled volume growth values isn't valid. The PNSD shape will change with aerosol type as will the Kd size distribution. Aerosol size-dependent growth varies with size such that multiplying an entire Kd distribution by a constant won't reproduce the Kd distribution of a different aerosol types. In addition the Kd and frh data come from two different measurement sites.

Please also note the supplement to this comment:
https://www.atmos-meas-tech-discuss.net/amt-2017-330/amt-2017-330-RC2-supplement.pdf

———————————————————

---

## Author Comment (AC1) · 23 Mar 2018

Thanks for your comments. The responses and the revised manuscript are included in the supplement zip file.

Please also note the supplement to this comment:
https://www.atmos-meas-tech-discuss.net/amt-2017-330/amt-2017-330-AC1-supplement.zip

---

## Author Response (AR1)

**Response to anonymous referee #2**

**General comments**:

The paper, "A novel method for calculating ambient aerosol liquid water contents based on measurements of a humidified nephelometer system" describes a new technique to measure aerosol liquid water content (LWC). The method described has advantages over traditional methods (ie HTDMA) as it measures LWC of aerosols in real time. In addition, this three wavelength nephelometer system measures the entire size distribution at once without assuming a constant growth factor for the entire distribution, as is the case with previous nephelometer measurements. The LWC of aerosols has important implications for climate forcing and atmospheric chemistry and there is a need for a more accurate measurement to reduce uncertainties.

**Response**: Thanks for your comments.

**Comment**: In general, there are large sections of this paper that should be omitted. The content does not support the conclusions of the paper. More importantly, these sections are sometimes confusing and will only distract the reader. This includes the paragraphs describing the relationship between scattering and aerosol volume, which the authors show to be correlated but not easily parameterized. The next section, which describes using the angstrom exponent and HBF to constrain the ratio between scattering and aerosol volume can also be summarized or omitted. The author's conclusion that large bias' will occur if using the "look up table" (fig 4) isn't necessary for the sake of the papers conclusions. More emphasis needs to be placed on what was done that produced usable results.

**Responses**: Thanks for your comments. We have revised the manuscript substantially. Now the paper has four sections. Section 1 is the introduction. Section 2 is instruments and datasets. Section 3 is methodology. Section 4 is results and discussions. The proposed method has two steps. The first step is calculating $V_a$ (dry) based on measurements of the "dry" nephelometer using a machine learning method called random forest model. The second step is calculating Vg(RH) based on Ångström exponent and $f$(RH) measured by the humidified nephelometer system. Therefore, Section 3.1 is the closure calculations between measured and calculated scattering coefficients, because the selection of datasets for training is crucial for machine learning method. In Section 3.2, the calculation of $V_a$(dry) using the random forest model is discussed. In Section 3.3, the method of calculating Vg(RH) based on Ångström exponent and $f$(RH) are presented and discussed. In Section 3.4, the formula of calculating ambient ALWC is described. In Section 4.1, the method of predicting $V_a$(dry) using the trained random forest model is validated by using measurements of Wangdu campaign. In Section 4.2, the calculated ambient ALWC using the proposed method of connecting $f$(RH) to Vg(RH) is verified with ambient ALWC calculated from ISORROPIA thermodynamic model. In Section 4.3, the volume fractions of water in ambient aerosols are described and discussed. In Section 4.4, the applicability of the proposed method for calculating ALWC is discussed.

**Comment**: Finally, the authors describe the machine learning method, which improves the ability to predict the volume of aerosol in the dry state. During the Wangdu campaign (Fig6), this method is able to reproduce the dry volume very well. My main concern is how applicable this method would be in a different type of environment. How difficult would it be to train the estimator to respond to different data sets and measurement conditions?

**Responses**: Thanks for your comment. Due to the lack of PNSD and nephelometer measurements under different environment conditions, the proposed machine learning method is not validated using measurements from different environment conditions. We cannot be sure if this trained estimator can be applicable under different environment conditions. Thus, we have recommended that the estimator should be trained with regional historical datasets. As what's shown in Fig.5, the training of this random forest model requires only datasets of simultaneously measured PNSD and BC which are already being measured for years in some regions.

**Comment**: The next section describes parametrizing the relationship between f(RH) and humidified aerosol volume using the "look up table" shown in fig 8. I would recommend referring to both fig 8 and fig 4 as something other than a lookup table, which is not an appropriate description for this plot. This approach, once again seems limited by the specific environment. It would be nice if the authors showed results from a different less-polluted region.

**Response**: Thanks for your comment. Based on your comment, results of Fig.4 are omitted. As to results of Fig.8 (Fig.6 in the revised manuscript), we have validated the way of connecting $f(\mathrm{RH})$ to $\mathrm{Vg(RH)}$ by using results in Fig.6 as a look up table with ambient ALWC calculated from ISORROPIA model. And we also have compared results of this method with the results of using the traditional way of calculating $\mathrm{Vg(RH)}$ based on $f(\mathrm{RH})$ (Guo et al., 2015). The results indicate that the proposed method can improve the calculation of $\mathrm{Vg(RH)}$ based on measured $f(\mathrm{RH})$.

**Comment**: The paper has multiple typographical and grammatical errors. Line 365 is a good example of the grammatical/typographical errors that are found throughout. Line 405 references a figure 20a, which doesn't exist. Prior to publication I would recommend careful editing for these errors as well as re-formatting to streamline the paper for only the most pertinent information.

**Response**: Thanks for your comment. We have edited the English with a copy editor and checked typographical and grammatical errors.

**References**

Guo, H., Xu, L., Bougiatioti, A., Cerully, K. M., Capps, S. L., Hite Jr, J. R., Carlton, A. G., Lee, S. H., Bergin, M. H., Ng, N. L., Nenes, A., and Weber, R. J.: Fine-particle water and pH in the southeastern United States, Atmos. Chem. Phys., 15, 5211-5228, 10.5194/acp-15-5211-2015, 2015.

**Response to anonymous referee #3**

**General Comments**:

The authors present both empirical and machine-learning methods for determining an aggregate or effective volume and the aerosol water content from the nephelometer Angstrom and backscattering coefficients. The empirical method uses cross correlations between the aerosol scattering coefficient, measured volume, backscatter fraction, and Angstrom exponent to estimate the aerosol volume. The machine learning method uses backscatter and Angstrom exponents to mimic the aerosol scattering to volume ratio. The machine-learning method offers a valuable tool that could be applied to many aspects of atmospheric aerosol and chemical predictions. The paper needs further development. I think it's important to refine the methodology, improve the paper organization, and clarify some of text to make this a stronger paper. The section linking scattering hygroscopic growth to volume hygroscopic growth doesn't follow a valid analysis method. The fRH and gRH data come from different measurement sites. I suggest leaving out the sections on volume hygroscopic growth as well as the discussion on kappa-Kohler. I don't recommend publication until the paper is restructured.

I suggest resubmitting the paper after removal of the sections on hygroscopic growth and water content.

**Responses**: Thanks for your comments and insightful suggestions. Based on your comments, we have refined the methodology and reorganized the paper. As to the section linking scattering hygroscopic growth to volume hygroscopic growth, this section is an important part of our methodology, and is moved to the methodology section. For the size-resolved κ distributions used for simulating the relationship between scattering hygroscopic growth to volume hygroscopic growth, only the average shape of the size-resolved κ from HaChi is used because that ratios range from 0.05 to 2 with an interval of 0.05 are multiplied with the average size-resolved κ distribution (the black line shown in Fig.7a) to produce a number of size-resolved κ distributions which represent aerosol particles from nearly hydrophobic to highly hygroscopic. Results from other studies have shown similar size dependence of aerosol hygroscopicity (Meng et al., 2014). We also have done the comparison between ambient ALWC (aerosol liquid water content) calculated from measurements of the humidified nephelometer system by using the proposed method and ambient ALWC calculated from ISORROPIA thermodynamic model. A good agreement is achieved between them. But if use the traditional way of connecting $f(\text{RH})$ (scattering enhancement factor) to Vg(RH) (volume growth factor), the ambient ALWC tends to be significantly overestimated, especially when RH is higher than 80%. Thus, we think this part provides a new way for connecting $f(\text{RH})$ to Vg(RH) which is useful for estimating ambient ALWC based on measurements of the humidified nephelometer system.

**Comment**: The paper needs better organization and clear, step-wise presentation of the methods and results. The methodology is scattered throughout the paper. Description of the results is vague and tends to gloss over important features.

**Response**: Thanks for your comment. We have revised the methodology part substantially. The method of estimating $V_a(\text{dry})$ using the machine learning method and the way of connecting $f(\text{RH})$ to $\text{Vg(RH)}$ are moved to the methodology section.

**Comment**: Please edit the English grammar and word order. Avoid run-on sentences. You have attendency to state conclusions without providing supporting evidence. State the methods used and then the data results. The methods and results are interspersed in the paper, which adds to confusion.

**Response**: Thanks for your comment. We have edited the English by another copy editor. In the revised manuscript, the methods and results are separated. The used datasets are introduced in Sect.2. Calculation method of $V_a(\text{dry})$ based only on measurements of the nephelometer is described in Sect.3.2. The way of deriving $\text{Vg(RH)}$ based on measurements of the humidified nephelometer system is introduced and discussed in Sect.3.3. The final formula of calculating ambient ALWC is described in Sect.3.4. The verification of the $V_a(\text{dry})$ predicted by using the machine learning method is described in Sect.4.1. The validation of ambient ALWC calculated from measurements of the humidified nephelometer system is presented in Sect.4.2. And the contribution of ambient ALWC to total ambient aerosol volume is discussed in Sect.4.3.

**Comment**: Introduction 1. Although the methodology is different from other inversion techniques, such as those of Ziegar et al., that calculate an "effective" gRH from fRH, the methodology is similar to that used in Aeronet retrievals of the aerosol effective radius from the AOD, Angstrom exponent and asymmetry parameter. There was a paper that attempted to calculate fRH or aerosol water using Aeronet data, but it suffered from low signal and spatial resolution. Can the authors describe how their method is similar to or different from the Aeronet retrievals and also speculate if this method could be used with remote sensing AOD measurements? Below is a link that has links to their "spectral deconvolution algorithm".

https://aeronet.gsfc.nasa.gov/new_web/Documents/Inversion_products_V2.pdf or Atmos. Meas. Tech., 10, 695-708, 2017, https://doi.org/10.5194/amt-10-695-2017.

**Response**: Thanks for your comment. We think our method is different with Aeronet retrievals. Our method is dealing with optical properties of aerosols at one location. Aeronet retrievals are dealing with integral optical properties of aerosols that distributed from the surface to the top of the atmosphere and assumes that aerosols are homogeneously distributed across the vertical layer. However, in real world, microphysical properties of aerosol particles (aerosol size distributions, aerosol hygroscopicity, aerosol mixing state, et al) at different altitudes are different, and relative humidity of the air at different altitudes also differs greatly (Kuang et al., 2016). Our method is based on machine learning which learn from historical datasets, and six parameters are used to constrain variations in PNSD in the machine learning method. The way of connecting $f$(RH) to Vg(RH) is based on simulative experiment which only assumes an average shape of size dependence of aerosol hygroscopicity and the variation of bulk aerosol hygroscopicity is considered. The Aeronet retrievals retrieves the particle number size distribution, complex refractive index and partition of spherical/non- spherical particles which fits the observed data best. We think the method proposed in this research can not be used with remote sensing AOD measurements, because it is difficult to use several parameters to constrain PNSD and RH variations at different altitudes, too much about aerosol properties and aerosol vertical distribution are unknown.

**Comment**: 2. The introduction needs to state more about the methodology other than saying it's "a novel method". Add a paragraph describing the two techniques. Describe the empirical model use of size-dependent parameters (backscatter and Angstrom) to predict the ratio of scattering/volume. Describe how machine-learning methods, using large data sets, over a long-time period as input, mimic the system behavior via feed-back loops to predict an output.

**Response**: Thanks for your comment. The following paragraph is added to the introduction: "In this paper, we propose a novel method to calculate the ALWC based only on measurements of a humidified nephelometer system.   The proposed method includes two steps. The first step is calculating $V_a(\text{dry})$ based on measurements of the "dry" nephelometer using a machine learning method called random forest model. With measurements of PNSD and BC, the six parameters measured by the nephelometer can be simulated using the Mie theory, and the $V_a(\text{dry})$ can also be calculated based on PNSD. Therefore, the random forest model can be trained with only regional historical datasets of PNSD and BC. The second step is calculating  Vg(RH) based on the Ångström exponent and $f(\text{RH})$ measured by the humidified nephelometer system. In this step, both the influences of the variations in PNSD and aerosol hygroscopicity are both taken into account to derive Vg(RH) from measured $f$(RH). Finally, based on calculated $V_a$(dry) and Vg(RH), ALWCs at different RH points can be estimated. The used datasets are introduced in Sect.2. Calculation method of $V_a$(dry) based only on measurements of the nephelometer, which measures optical properties of aerosols in dry state, is described in Sect.3.2. The way of deriving Vg(RH) based on measurements of the humidified nephelometer system is introduced and discussed in Sect.3.3. The final formula of calculating ambient ALWC is described in Sect.3.4. The verification of the $V_a$(dry) predicted by using the machine learning method is described in Sect.4.1. The validation of ambient ALWC calculated from measurements of the humidified nephelometer system is presented in Sect.4.2. The contribution of ambient ALWC to the total ambient aerosol volume is discussed in Sect.4.3. ". The random forest model is introduced in the methodology section.

**Comment**: 3. In Section 2.1 Transfer Table S1 to Section 2.1. You can leave the detailed measurement description in the supplement. Give general information on the breakdown of the aerosol composition between organics/sulfate/nitrate/dust at this time of year. Sections 2.2 to 3.2 need to be reorganized. I suggest segregating Section 2 into Section 2.2 is closure, 2.3 is Mie theory, 2.4 Machine learning method.

I suggest renaming section 2.2 as "Closure Calculations". Show the scattering closure between the measured and calculated scattering coefficient. The integrity of the volume and fRH closure depends on the scattering closure. Figure 2 would validate the measurements better if 2a showed the scattering closure and 2b showed the scattering vs volume.

**Response**: Thanks for your comment. Section 2 is about the instruments and datasets. Table S1 is transferred to this section, more details are listed in this Table. Section 2 and Section 3 are reorganized. The closure results between the measured and calculated scattering coefficient during different campaigns are introduced in Section 2.1. The discussions about theoretical relationships between scattering coefficient and aerosol volume are introduced and discussed in Section 2.2.1. The machine learning method is introduced in Section 2.2.2. The proposed method of connecting $f(\text{RH})$ to Vg(RH) is introduced in Section 3.

**Comment**: Figure 2 shows 2 branches or subsets of the data; one above and a 2nd below the fit line. Is this behavior present in the scattering closure? Do these two branches represent 2 different aerosol types or multiple size modes?

**Response**: Thanks for your comment. These two branches exist for datasets of campaign F1 (please refer to Table 2 of the revised manuscript for detailed campaign information). As shown in Fig.1 of the revised manuscript. This behavior does not present in the scattering closure. The relationship between $V_a(dry)$ and $\sigma_{sp}$ for measurements of

[Figure]

**Figure 1**. (a) The relationship between $V_a(dry)$ and $\sigma_{sp}$ at 550 nm for measurements of campaign F1. (b) The blue one corresponds to average PNSD of data points of lower branch which locate in the range of dashed blue lines in (a), red one corresponds to average PNSD of data points of upper branch which locate in the range of dashed red lines in (a).

campaign F1 is shown in Figure 1a. And the average PNSDs of chosen data points of lower and upper branches are shown in Fig.1b. These results indicate that two branches corresponding two different PNSD shapes, but without multiple size modes.

**Comment**: The application of an average Rvsp to estimate the volume doesn't add to the paper and distracts from the other methods. I suggest removing it from the analysis.

**Response**: Thanks for your comment. This part is removed from the manuscript.

**Comment**: Section 2.3 Mie Move equations 5 and 6 to the start of section 2.3 and explain how you relate scattering to volume and the assumptions in this approach. Move the discussion of Mie theory from section 3.1 to section 2.3. State your adaptation of

Mie Theory in a clear, stepwise, logical fashion. Show that going from equations 5 >

>6 assumes that Q is roughly linear with r such that Q=k*Q(m).

Describe Mie model using simulated data with 4 aerosol types and results in Figure 3.

Describe limitations of assumption that Q is linearly proportional to r.

Describe what the dotted red lines represent in the Figure 3 caption.

**Response**: Thanks for your comment. We have moved equations 5 and 6 to the section 3.2. The name of this section is "calculation of $V_a(\text{dry})$ based on measurements of the "dry" nephelometer". In Sect.3.2.1, we have described the theoretical relationship between $V_a(\text{dry})$ and $\sigma_{sp}$. In Sect.3.2.2, we have described the machine learning method. The discussion of the Mie theory is also moved to Sect.3.2. Limitations of the assumption that Q is linearly proportional to r is discussed in Sect.3.2.1. The dotted red lines in Figure 3 (Figure 2 of the revised manuscript) are described.

**Comment**: Section 2.5 Machine learning

Describing an alternate method of machine learning using size-dependent scattering parameters; Angstrom and backscatter to mimic the measured scattering/volume ratio. Give some background on "machine learning" and the algorithm name. Can you add a simplified algorithm decision tree with basic logic steps or diagram that would help explain the process?

**Response**: Thanks for your comment. The background on machine learning and the algorithm name is described in Sect.3.2.2. A schematic diagram of training the machine learning method is also shown in Figure 5.

**Comment**: Results and Discussion: Section 3.1 Empirical method Refer back to Figure and need for estimating a variable kscat or R. Introduce the empirical method of determining R from HBF and Angstrom exponent. Explain figure 4.

In your description of the results, use and simple and direct language. Leave out extra information on the impact of BC and mixing state on the HBF and Angstrom exponents until the discussion of the data fit lines as these are secondary contributions.

Explain Figure 5a and variance about fit line in relation to HBF. This variance likely stems from the Angstrom exponent and HBF describing a fraction of the PNSD. Show a plot of HBF(450, 550, 700nm) and Angstrom exponents(450/550, 450/700 and 550/700) versus r for a lognormal size distribution. The plot will show the sensitivities of these parameters to aerosol size.

**Response**: Thanks for your comment. Comments from another reviewer suggest that results of Figure 4 and Figure 5a should be omitted. We agree with reviewer and focus on the machine learning method. HBF(450, 550, 700nm) and Angstrom exponents (450/550, 450/700 and 550/700) as a function of particle diameters are shown in Fig.4 of the revised manuscript. The results shown in Fig.4 indicate that HBFs at three wavelengths and Ångström exponents calculated from $\sigma_{sp}$ at different wavelengths are sensitive to different diameter ranges of PNSD.

**Comment**: Section 3.2 Machine learning Explain how machine method uses 6 parameters to describe the aerosol volume relative to the scattering. Explain figures 5b and Figure 6 and how machine learning is an improvement over the empirical method.

**Response**: Thanks for your comment. A schematic diagram of training the machine learning method is shown in Figure 5. Why machine learning is an improvement is explained in Sect.3.2.2.

**Comment**: 3.3 fRH and Vrh

The method of simulating the Kd size distribution from variations of the average and then applying this to the measured size distribution to obtain a 4 modeled volume growth values isn't valid. The PNSD shape will change with aerosol type as will the Kd size distribution. Aerosol size-dependent growth varies with size such that multiplying an entire Kd distribution by a constant won't reproduce the Kd distribution of a different aerosol types. In addition the Kd and frh data come from two different measurement sites.

**Response**: Thanks for your comment. Section 3.3 is an important part of our methodology, and is moved to the methodology section of the revised manuscript. For the size-resolved $\kappa$ distributions used for simulating the relationship between scattering hygroscopic growth to volume hygroscopic growth, only the average shape of the size-resolved $\kappa$ from HaChi is used because that ratios range from 0.05 to 2 with an interval of 0.05 are multiplied with the average size-resolved $\kappa$ distribution (the black line shown in Fig.7a) to produce a number of size-resolved $\kappa$ distributions which represent aerosol particles from nearly hydrophobic to highly hygroscopic. We agree with the reviewer that PNSD shape as well as size-resolved $\kappa$ distribution will change. However, $f(RH)$ and $Vg(RH)$ are integral variables which are sensitive to integral variables which can represents variations in PNSD and overall aerosol hygroscopicity. This is why we establish a look up table which can take the variations of bulk aerosol hygroscopicity and the Ångström exponent into account. We also have examined how much the variations in shape of size-resolved $\kappa$ distribution and PNSD will impact on the prediction ability of the established look up table based on measure size-resolved $\kappa$ distributions. The results are shown in Figure 7b. Moreover, results from other studies have also shown similar size dependence of aerosol hygroscopicity (Meng et al., 2014). We also have done the comparison between ambient ALWC (aerosol liquid water content) calculated from measurements of the humidified nephelometer system by using the proposed method and ambient ALWC calculated from ISORROPIA thermodynamic model. A good agreement is achieved between them. A traditional way of connecting $f(\text{RH})$ to $\text{Vg}(\text{RH})$ (Guo et al., 2015) is also described and discussed in Sect.4.3. If we use the traditional way of connecting $f(\text{RH})$ to $\text{Vg}(\text{RH})$, the ambient ALWC tends to be overestimated significantly, especially when RH is higher than 80%. Thus, we think this part provides a new way of connecting $f(\text{RH})$ to $\text{Vg}(\text{RH})$ which is useful for estimating 
[revised manuscript text omitted]

---

## Author Response (AR2)

**Dear Editor:**

Thank you and all the reviewers for the quite constructive and helpful comments! All these comments raised by the referees have been explicitly replied point by point and incorporated into the revision. All authors consent to the revisions and the responses.

Thank you very much for your attention and consideration.

Sincerely Yours

Chunsheng Zhao

**Responses to anonymous referee #1**

**Comment**: 1. Lines 265-271: I am confused which dataset the authors used for training the random forest model. It seems that the authors used the measurements of TSI 3563, however, the campaign F6 used Aurora 3000 (table 2). Meanwhile, the authors claimed that good agreement between measured  $\sigma sp$  and that calculated based on measured PNSD and BC with Mie theory was found in the campaigns F1, F4, F5 and F6 (section 3.1). so why you use dataset from field campaigns F1 to F4 and F6. Please check and explain.

**Response**: Thanks for your comment. To avoid that the measurements uncertainties are involved in the training processes of the random forest model. Datasets of PNSD and BC from field campaigns F1 to F4 and F6 are used to calculate  $V_a(dry)$  and simulate six optical parameters corresponding to measurements of TSI 3563. Calculated  $V_a(dry)$  and simulated six optical properties are then used as inputs for training the random forest model. This process is independent of simultaneously measured aerosol optical properties from the nephelometer, and therefore not relevant with what nephelometer has been used. The reason that datasets of PNSD and BC from field campaign F6 are also used, is that we want the trained random forest model can cover different aerosol loadings as much as possible. Also, the closure results between modelled and measured  $\sigma_{sp}$  do not affect whether the PNSD and BC should be used in the training process. However, the closure results are important for deciding if the simultaneously measured PNSD and aerosol optical properties (scattering and backscattering coefficients) should be used for validating the machine learning model.

**Comment**: 2. Figure 5. This figure shows the schematic diagram of step 1 of the proposed ALWC calculation method. Is it possible to add the information of step 2 in this figure?

**Response**: Thanks for your comment. This figure shows how to train the random forest model. It is difficult to add the information of step 2 in this figure. However, we think it is important to show the flowchart of calculating the ambient aerosol liquid water contents based on measurements of the humidified nephelometer system and this flowchart is shown in Fig.8 of the revised manuscript.

Comment: Line 192: diameter

**Response**: Thanks for your comment, we have revised this word.

**Comment**: Line 196: equation (5)? I think it should be equation (1).

**Response**: Thanks for your comment. We have revised equation (5) to equation (1)

**Comment:** Line 344: Figure 7 came first than figure 6.

**Response:** Thanks for your comment. This problem is solved by adding a figure in the supplement which describes the used average size-resolved  $\kappa$  distribution.

Comment: Figure 8: please add units.

**Response**: Thanks for your comment. The unit of  $V_a(dry)$  is added in the figure caption.

Figure 9: I think the unit of ALWC is wrong, please revise.

**Response**: Thanks for your comment. The unit of ALWC is revised.

**Responses to anonymous referee #2**

**Comment**: Dust is pervasive in the NCP. If not from the Loess regions, then roads and construction dust are present. This may be a factor influencing differences between the PM2.5 and PM10 aerosol in this study.

**Response**: Thanks for your comment. In Sect.4.4 of the revised manuscript, we emphasized that cautions should be exercised if using the proposed method to estimate the ALWC when the air mass is significantly influenced by sea salt or dust.

**Comment**: You need to stress in the Introduction that this study uses data from multiple sites to characterize a regional aerosol. Otherwise, mixing data from multiple sites over different time periods would give erroneous results. The quantity of data from multiple sites used in the machine learning code is large enough to infer the results apply to a larger data population and region.

**Response**: Thanks for your comment. The following sentence is added in the last paragraph of the introduction of revised manuscript: "In this study, datasets of PNSD and BC measured from multiple sites are used in the machine learning model to characterize a regional aerosol and these datasets have covered a wide range of aerosol loadings."

Comment: Title: Change "contents" to "content"

**Response**: Thanks for your comment. We have revised the title accordingly.

Comment: Page1: Change "so far" to "before now"

**Response**: Thanks for your comment. We have revised the abstract accordingly.

**Comment**: Page 9, line 182: Did you mean "either or" instead of "neither nor"? **Response**: Thanks for your comment. We mean "either or", and the manuscript is revised accordingly.

**Comment**: Line 183... view of this, Qscat at 550 nm, as a function of particle diameter for four types of aerosol particles, is simulated...

**Response**: Thanks for your comment. We have revised the manuscript accordingly.

Comment: Page 15, line 300: "Kelvin"

**Response**: Thanks for your comment. We have revised this word accordingly!

**Comment**: Lines 302-304: You show in Figure 7a a large size-dependence to k with aerosol size. Use of a single k volume parameter doesn't assume a constant k value with particles size. Rather it implies that a kappa(volume) can be expressed as a single value that perhaps is proportional to a weighted average of the size-dependent kappa (diameter) values. Such that  $k \ vol = w \ k(D)$ . This assumption allows Vg(RH) to be expressed as a linear function.

**Response**: Thanks for your comment. The sentence "which means that if  $\kappa$  values of aerosol particles of different sizes are the same" is rewrote as "If a constant  $\kappa$  which

represents the overall aerosol hygroscopicity of ambient aerosol particles, is used as the  $\kappa$  of different particle sizes"

Comment: Page 16, line324: replace "consolidate" with "validate"

**Response**: Thanks for your comment. We have revised the manuscript accordingly.

**Comment**: Page 18, lines364-366: rewrite "Figure 6 shows the influence of aerosol size and chemistry on Rvf. For Angstrom exponents less than ~1.1, Rvf varies strongly with ksca. However for Angstrom exponent values greater than ~1.1, the Rvf relative standard deviation exhibits a higher variability with the Angstrom exponent; thus showing the sensitivity of Rvf to changes in aerosol size for small particles."

**Response**: Thanks for your comment. This part is revised as "Simulated values of  $R_{Vf}$  range from 0.8 to 1.7 with an average of 1.2. Overall,  $R_{Vf}$  value is lower when value of Ångström exponent is larger. The percentile value of standard deviation of  $R_{Vf}$  values within each grid divided by its average is shown in Fig.6b. In most cases, these percentile values are less than 10% (about 90%) which demonstrates that  $R_{Vf}$  varies little within each grid shown in Fig.6a. Figure 6 shows the influence of aerosol size and chemistry on  $R_{Vf}$ . For Ångström exponent less than ~1.1,  $R_{Vf}$  varies strongly with  $\kappa_{sca}$ . However, for Ångström exponent values greater than ~1.1, the  $R_{Vf}$  relative standard deviation exhibits a higher variability with the Ångström exponent. Thus, showing the sensitivity of  $R_{Vf}$  to changes in aerosol size for small particles. In general, results shown in Fig.6 imply that results of Fig.6a can serve as a look up

table to estimate  $R_{Vf}$  and thereby  $\kappa_{Vf}$ , such that these values can be directly predicted from measurements of a three-wavelength humidified nephelometer system."

**Comment**: Line 370:rewrite " ...thereby kf, such that these values can be directly..." As a note, the high variability of Rvf at high Angstrom exponents may result from differences between monomodal and bimodal size distributions.

Response: Thanks for your comment. We have revised the manuscript accordingly.

**Comment**: Line 379: rewrite "... is highly variable yet has no apparent correlation with aerosol loading."

**Response**: Thanks for your comment. We have revised the manuscript accordingly. **Comment**: Page 23, line 478: rewrite "On average, when ambient …"

**Response**: Thanks for your comment. We have revised the manuscript accordingly.